# Selective and noncovalent targeting of RAS mutants for inhibition and degradation

Kai Wen Teng[1], Steven T. Tsai[1], Takamitsu Hattori[1,2], Carmine Fedele[1], Akiko Koide[1,3], Chao Yang[4], Xuben Hou[4], Yingkai Zhang [4], Benjamin G. Neel[1,3], John P. O'Bryan[5,6] & Shohei Koide [1,2✉]

Activating mutants of *RAS* are commonly found in human cancers, but to date selective targeting of RAS in the clinic has been limited to KRAS(G12C) through covalent inhibitors. Here, we report a monobody, termed 12VC1, that recognizes the active state of both KRAS (G12V) and KRAS(G12C) up to 400-times more tightly than wild-type KRAS. The crystal structures reveal that 12VC1 recognizes the mutations through a shallow pocket, and 12VC1 competes against RAS-effector interaction. When expressed intracellularly, 12VC1 potently inhibits ERK activation and the proliferation of RAS-driven cancer cell lines in vitro and in mouse xenograft models. 12VC1 fused to VHL selectively degrades the KRAS mutants and provides more extended suppression of mutant RAS activity than inhibition by 12VC1 alone. These results demonstrate the feasibility of selective targeting and degradation of KRAS mutants in the active state with noncovalent reagents and provide a starting point for designing noncovalent therapeutics against oncogenic RAS mutants.

[1] Perlmutter Cancer Center, New York University Langone Health, New York, NY, USA. [2] Department of Biochemistry and Molecular Pharmacology, New York University Grossman School of Medicine, New York, NY, USA. [3] Department of Medicine, New York University Grossman School of Medicine, New York, NY, USA. [4] Department of Chemistry, New York University, New York, NY, USA. [5] Department of Cell and Molecular Pharmacology and Experimental Therapeutics, Medical University of South Carolina, Charleston, SC, USA. [6] Ralph H. Johnson VA Medical Center, Charleston, SC, USA. ✉email: Shohei.Koide@nyulangone.org

Activating mutations in the *RAS* genes are frequently present in human tumors, and these *RAS* mutants play important roles in oncogenic transformation[1,2]. Genetic knockdown or silencing approaches have established that directly targeting *RAS* mutants is effective in inhibiting *RAS*-driven cancers[3,4]. However, these approaches suppressed the expression of *KRAS* mutant genes, which does not fully mimic the inhibition of KRAS proteins by a drug. Recent clinical trials of G12C allele-specific inhibitors revealed high dose tolerance and effective tumor reduction in certain patients, supporting selective inhibition of RAS mutants as a viable therapeutic strategy against KRAS (G12C)-driven cancer[5–7]. By contrast, despite extensive effort, potent and selective inhibitors against other RAS mutants are still lacking. Consequently, we still do not know whether or not it is feasible to develop such inhibitors of any molecular class. Most oncogenic RAS mutants do not have a unique chemically reactive group suitable for selective targeting with covalent inhibitors. In addition, the covalent inhibitors bind to an area under the switch II region, called S-II pocket, that is present in the GDP-bound state and form covalent linkage to Cys12[8,9]. These G12C-selective inhibitors are effective, because KRAS(G12C) cycles intrinsically between the active, GTP-bound form and the inactive, GDP-bound form, and the compounds lock KRAS(G12C) in the inactive state[8]. Most mutants, such as KRAS(G12V) that accounts for up to 30% of RAS mutations in certain tumor types[1,7], have slow intrinsic nucleotide exchange rates and thus remain in the active state for an extended period[10]. Thus, targeting the active state should be the preferred approach for many RAS mutants. However, one can also argue that targeting the active state may present an additional challenge, because an inhibitor needs to effectively compete against multiple RAS-binding effectors that also bind to the active state[11]. Due to the lack of suitable inhibitors, it remains unclear whether a selective, noncovalent inhibitor targeting the active state of an oncogenic RAS mutant is effective in suppressing oncogenic RAS-mediated signaling and tumor growth.

The absence of selective inhibitors for most RAS mutants strongly suggest that developing such inhibitors requires a different approach. To develop drug-like molecules for proof-of-concept purposes, many binding proteins targeting the GTP-bound state of RAS mutants have been developed with a hope that larger binding surfaces of proteins coupled with very large sequence diversity afforded by molecular display technologies could achieve high selectivity[12–16]. Unfortunately, these RAS-binding proteins reported to date are either not selective for mutants (over wild-type (WT) KRAS) or not effective in inhibiting RAS-mediated signaling in cells. These molecules may have insufficient selectivity to effectively engage RAS mutants in cells where potentially excess concentrations of wild-type RAS isoforms serve as a sink for low-specificity inhibitors. Alternatively, their lack of efficacy might reflect the challenge of competing against effectors[11]. To conclusively determine the effectiveness of directly and noncovalently targeting RAS mutants, one will require inhibitors with very high selectivity and affinity.

To develop highly selective inhibitors against oncogenic mutants, we employed the monobody technology, a synthetic binding protein platform derived from a human fibronectin type III domain[17,18], a small domain in the immunoglobulin superfamily. The monobody platform has produced potent and selective inhibitors to diverse protein targets, such as monobodies that selectively bind to a single member among ~120 human SH2 domains[19]. The monobody platform is ideally suited for developing tool inhibitors against intracellular targets, because one can deliver monobodies as genetically encoded reagents, as their lack of disulfide bonds, unlike conventional antibody fragments, makes it easy to express them in the functional form under the reducing environment of the cytoplasm. In a previous application to RAS, we have developed a monobody called NS1 that inhibits RAS-mediated signaling through targeting the α4–α5 surface, which has established an approach to control RAS functions[20].

Another important unanswered question in RAS drug discovery is whether or not one can selectively degrade an endogenous RAS mutant in a noncovalent manner. Proteolysis targeting chimeras (PROTACs) are an emerging drug modality, bi-functional molecules that direct a protein of interest (POI) to the E3 ligase for ubiquitination and subsequent degradation by the proteasome[21,22]. PROTACs can undergo multiple cycles of target engagement followed by degradation and should offer advantage over inhibition by occupancy[23,24]. However, due to a lack of suitable noncovalent warheads, no PROTAC molecules selectively targeting endogenous RAS mutants, except for covalent PROTACs for KRAS(G12C)[25], have been developed, making it difficult to determine the value of degradation-based therapy in RAS drug discovery. The covalent PROTACs targeting KRAS (G12C) requires one-to-one stoichiometry for target engagement, which excludes the possibility of multiple cycles of target degradation, an important aspect contributing to the effectiveness of PROTACs. We note that covalent PROTACs may still offer advantages over covalent inhibitors, because covalent PROTACs deplete the cells of KRAS mutants and eliminate potential scaffolding roles that the mutants may play. PROTAC-like degraders have also been developed by fusing the VHL subunit of the E3 ligase complex with a RAS-binding protein including KRAS-specific DARPin and the NS1 monobody[26,27]. However, these degraders were not mutant-selective and showed variation in efficacy. Notably, the KRAS-specific DARPin fused to VHL effectively inhibited the proliferation of RAS-driven cancer cell lines with minimal effects on selected WT-KRAS cell lines[26], suggesting the potential utility of KRAS-selective but not mutant-selective degraders. Taken together, there remain many challenges in the development of degraders against KRAS mutants.

Here, we report the successful development of a noncovalent inhibitor of RAS mutants with high selectivity toward the active, GTP-bound state of two oncogenic RAS mutants, KRAS(G12V) and KRAS(G12C). This monobody potently inhibited RAS-mediated signaling and demonstrated efficacy in both in vitro and in vivo models. We developed degradation-resistant PROTAC-like fusion proteins based on this monobody and demonstrated that they selectively and effectively degraded the RAS mutants. We found an optimal affinity window for efficient RAS degradation. Furthermore, we show that the RAS degrader suppressed the increase in RAS level observed in response to selective inhibition, and potently inhibited KRAS(G12C)-driven tumor growth in a mouse xenograft model. Taken together, using monobodies as "tool biologics", our results address a number of fundamental questions in RAS drug discovery and degrader development.

## Results

**A highly selective monobody against multiple RAS mutants**. We developed a monobody, termed 12VC1, that demonstrated high selectivity for KRAS(G12V) and KRAS(G12C) (Fig. 1, Supplementary Table 1) using an established method that combined phage display and yeast display technologies[18,20]. Binding measurements in the yeast display format revealed that 12VC1 bound selectively to the GTPγS-bound form, over the GDP-bound form, of the KRAS mutants, and, importantly, it did not detectably bind to wild-type K, N, or HRAS in either nucleotide-bound form (Fig. 1a, Supplementary Fig. 1a). Bio-layer Interferometry (BLI) using purified 12VC1 (Fig. 1b, Supplementary Table 2, Supplementary Fig. 1b) further demonstrated the high selectivity of 12VC1, with up to a 400-fold difference in affinity,

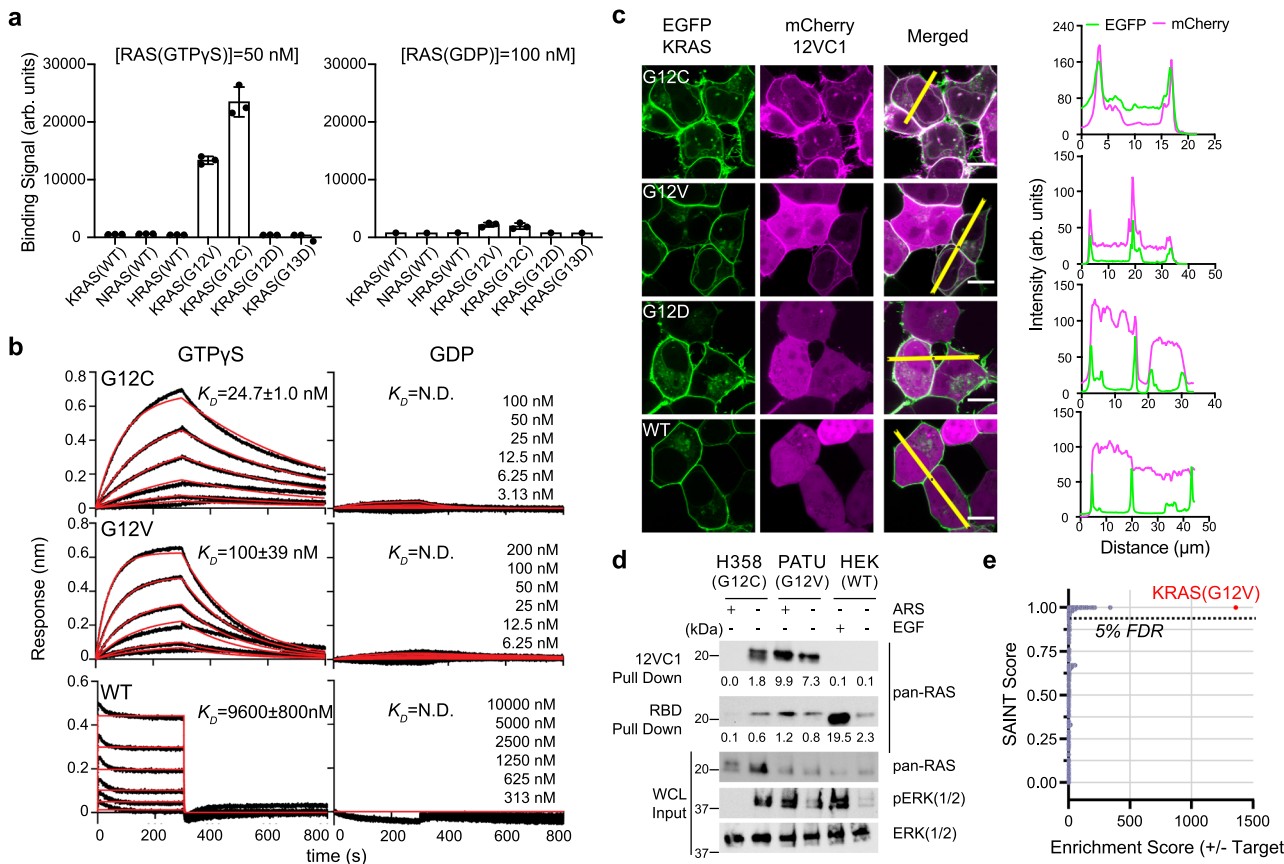

**Fig. 1 Monobody 12VC1 is selective to the G12C and G12V RAS mutants. a** Yeast display binding measurements of 12VC1 to the wild-type RAS isoforms and KRAS mutants in GTPγS (50 nM) or GDP (100 nM) bound forms. Each data point represents one technical replicate. Binding of biotinylated RAS was detected with neutravidin-Dylight 650. Binding signals in arbitrary units (arb. units.) are the median fluorescence intensity of the fluorescent population in the 75–95th percentile of the Dylight 650 detection channel of a flow cytometer. The mean and SD of three technical replicates are plotted. **b** BLI sensorgrams of 12VC1 binding against different KRAS mutants and WT in GTPγS or GDP-loaded state. The experimental BLI traces (black) for G12C and G12V in the GTPγS-bound state were globally fitted (red) using the 1:1 binding kinetic model. Steady-state global analysis was performed for KRAS(WT). The $K_D$ values shown are the mean ± SD from $n = 3$, technical replicates. N.D., not determined due to too weak binding. **c** Colocalization of mCherry fused 12VC1 (pseudo-color magenta) with overexpressed EGFP fused KRAS(WT) and mutants (pseudo-color green). Scale bar denotes 10 μm. The graphs on the right show the fluorescence intensity profiles under the yellow lines across the microscopy images. **d** Pull-down assay with biotinylated 12VC1 and GST-RBD of lysates of cell lines containing KRAS(G12C) and (G12V) and (WT) with and without ARS1620 and EGF treatments. Captured proteins (12VC1 pull down and RBD pull down) were probed using immunoblotting with a pan-RAS antibody. The amounts of RAS, total ERK, and pERK in the input lysate (WCL) were assessed using immunoblotting. Representative data shown (technical replicate, $n = 2$). **e** Affinity purified-mass spectrometry (AP-MS) analysis using 12VC1 as a capture reagent. The SAINT score[50] and enrichment score of each protein that was uniquely present in the affinity purified sample of KRAS(G12V) harboring cell line (PATU8902) over non-KRAS mutant cell line (A375) are plotted. The data showed that 12VC1 captured overwhelmingly more KRAS(G12V) from PATU8902 than from A375. The red dot represents SAINT score and enrichment score of KRAS(G12V). The dashed line signifies the cutoff for 5% false detection rate (FDR).

to mutants with certain small side chains over the wild type (Affinity: G12C > G12V >> G12A and G12S >> G12). These mutations collectively account for a large fraction of the *KRAS* mutations found in cancers with poor 5-year survival rates, including non-small cell lung cancer (NSCLC), colorectal cancer (CRC), and pancreatic ductal adenocarcinoma (PDAC)[1].

12VC1, when expressed intracellularly as a genetically encoded reagent, selectively engaged overexpressed KRAS mutants in the form of an EGFP fusion protein (Fig. 1c). A pull-down assay further confirmed its selectivity towards the GTP-bound state of RAS mutants fused to EGFP (Supplementary Fig. 2a). 12VC1 selectively captured endogenous KRAS from cancer cell lines PATU8902 and H358 that contain KRAS(G12V) and KRAS (G12C), respectively, but not from growth factor stimulated HEK293T cells containing only wild-type RAS (Fig. 1d). Furthermore, treatment of H358 cells with ARS1620, a covalent inhibitor that traps KRAS(G12C) in the GDP-bound state

abrogated detectable binding of RAS to 12VC1 (Fig. 1d, Supplementary Fig. 2b). A proteomics analysis showed that 12VC1 selectively captured endogenous KRAS(G12V) from PATU8902 cell lysates over wild-type RAS (Fig. 1e, Supplementary Fig. 2c–e). These results conclusively showed high selectivity of 12VC1 to both G12V and G12C mutants.

12VC1 competed with RAS-binding domain of RAF-1 (RAF1-RBD) for binding to RAS in a biochemical assay (Supplementary Fig. 3a). When expressed as a genetically encoded reagent, it potently inhibited RAS-mediated signaling in HEK293 cells overexpressing EGFP-KRAS(G12V) and -KRAS(G12C) but not EGFP-KRAS(G12D) (Supplementary Fig. 3b). Together, these results demonstrate that 12VC1 is a highly selective, noncovalent inhibitor of KRAS(G12V) and KRAS(G12C).

**Structural basis for mutant selectivity.** We determined the crystal structure of 12VC1 bound to HRAS(G12C) with GTPγS

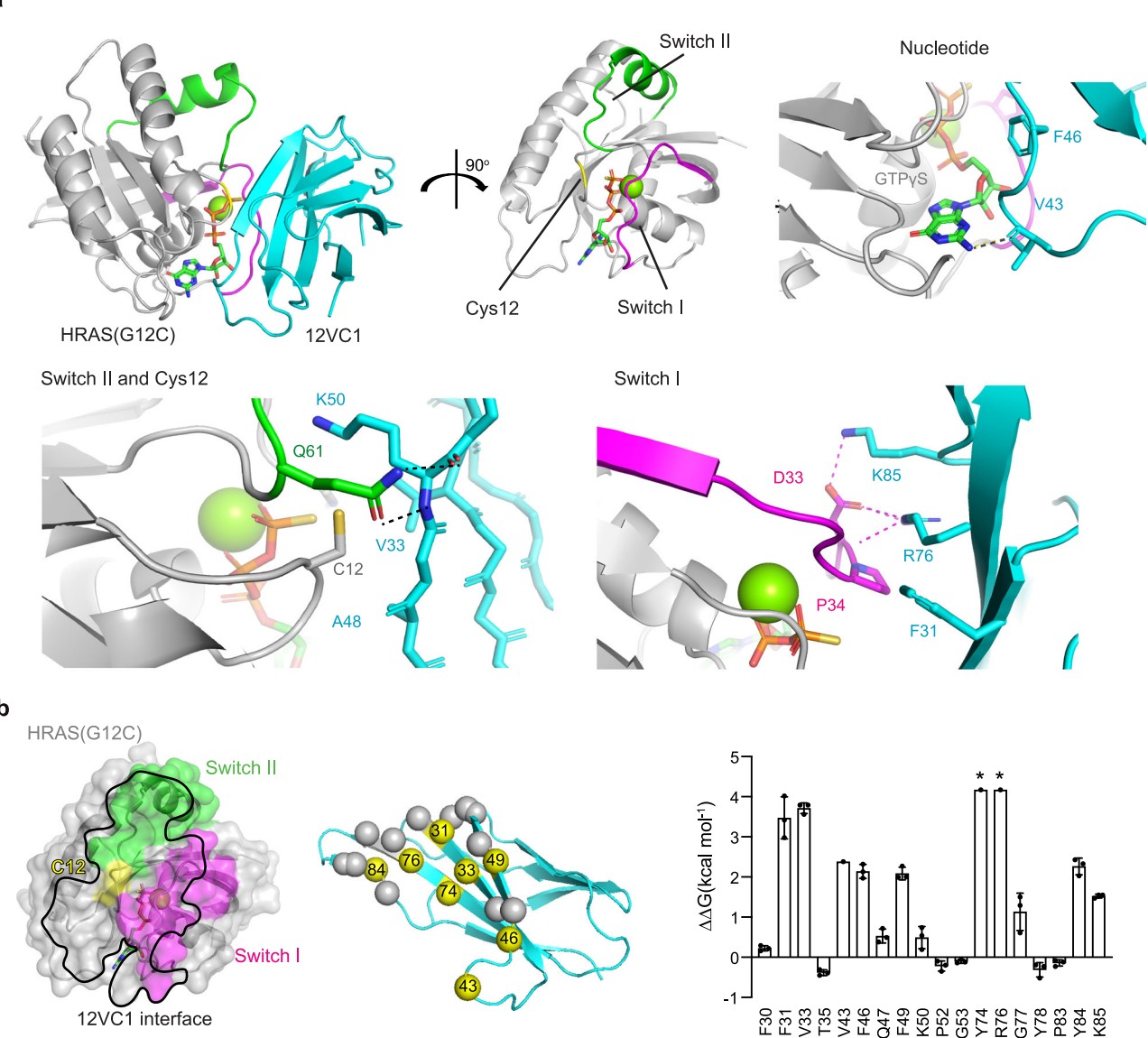

**Fig. 2 Structural basis for mutant-selective recognition of RAS by 12VC1 monobody. a** Crystal structure of 12VC1 (cyan) in complex with HRAS(G12C) bound to GTPγS (gray) at 2.54 Å resolution (PDB: 7L0G). Critical interactions between 12VC1 and HRAS(G12C) occur at the switch I (magenta) and switch II (green) regions. Interactions at these regions are expanded for detailed views in separate panels. Residue V43 of 12VC1 forms a hydrogen bound with the nucleotide. Mutated position, Cys12 of RAS (yellow), is accommodated by a pocket that consists of residues V33, A48, and K50 of 12VC1 shown in stick model. Residues R76 and K85 on 12VC1 form hydrogen bond and salt bridges with Switch I residue D33. Residue F31 on 12VC1 forms hydrophobic interactions with Switch I. **b** An open book view of the HRAS(G12C):12VC1 complex. Effects of alanine mutation of 12VC1 residues located within 4 Å of HRAS are shown in the bar graph (mean ± SD). Asterisks denote that ΔΔG is beyond the measurable limit of experiments and thus the values shown represent the lower limit. Experiment was performed in triplicate except for V43, Y74, and R76, and the error bars indicate s.d. The mutated residues in alanine scanning are shown as spheres in the cartoon model, and those for critical residues (ΔΔG > 2 kcal/mol) are colored yellow and labeled (middle panel).

(Supplementary Table 3) at 2.54 Å resolution. We used HRAS because it crystallizes more readily than KRAS[8]. The binding interface of this structure should be relevant to how 12VC1 recognizes KRAS mutants, because HRAS and KRAS have identical amino acid sequences in the effector lobe to which 12VC1 binds. Our binding data confirmed that 12VC1 maintained selectivity toward the G12C mutation in HRAS (Supplementary Fig. 4a). The crystal structure revealed that 12VC1 bound to RAS through both Switch I and II regions, as well as through the bound nucleotide, occupying 853 Å$^2$ of the surface of RAS (Fig. 2a; Supplementary Table 3). A comparison of the epitopes between HRAS bound to 12VC1 and RAF1-RBD showed an overlap in the Switch I region (Supplementary

Fig. 4b), which further confirmed that 12VC1 is a direct competitor of RAF1-RBD. The selectivity of 12VC1 to the active, GTP-bound form of RAS is likely due to the interactions of several residues of 12VC1 with residues of Switch I (Fig. 2a). Interestingly, Cys12 is the only residue in the P-loop that is in direct contact (within 4 Å) with 12VC1. Alanine scanning analysis of 12VC1 revealed numerous binding hot-spot residues distributed across the binding interface including those located in beta-strands and two loops (Fig. 2b, Supplementary Fig. 4c).

The structure and the interface energetics suggested two possible mechanisms for the high selectivity of 12VC1: first, the monobody directly discriminates the mutated residue at position 12, and second it recognizes a conformation of the epitope that is

unique to RAS mutants. To examine the first possibility, we utilized a computational structural analysis method, AlphaSpace[28], which revealed a shallow pocket, comprising residues V33, A48, and K50, that accommodates the Cys12 side chain (Fig. 2a, Supplementary Fig. 5a). This observation suggested that this pocket directly recognizes certain uncharged side chains at residue 12. Small differences in the side chain size and/or chemistry may substantially affect the interaction. The weaker affinity of 12VC1 to Gly, Ala, and Ser at position 12 (Fig. 1b, Supplementary Fig. 1b) can be rationalized in terms of penalty for not filling the pocket and not satisfying shape complementarity. However, the side chain volume is not the sole determinant of the binding affinity of 12VC1 to RAS mutants, as expected. In addition to being significantly larger than the Ser side chain (~30 versus ~50 × 10$^{-24}$ cm$^3$)[29], the Cys side chain is more hydrophobic, making it more energetically favorable to be buried in the pocket. Although Asp and Cys have similar side chain volumes[29], 12VC1 binds only marginally to KRAS(G12D) (Fig. 1a, Supplementary Fig. 1a), which can be rationalized by a large desolvation penalty associated with burying the Asp side chain that is much more electronegative than the Cys side chain. These considerations support the dominant role of direct recognition of the side chain at position 12 in achieving high selectivity.

A comparison of the 12VC1–HRAS(G12C) structure with the structure of 12VC1 bound to HRAS(WT) would allow us to conclusively differentiate the two possibility. However, we did not obtain crystals of the 12VC1–HRAS(WT) complex, probably due to its low-affinity interaction. Thus, we improved the affinity of 12VC1 to the wild type by mutating the residues that formed the pocket that "sensed" G12C (V33, A48, and K50; Fig. 2a) and by incorporating the T35A mutation that improved the overall affinity to RAS (Fig. 2b). The resulting clone, termed 12VC3, had improved affinity to wild-type RAS (Supplementary Table 1, Supplementary Fig. 5b, c), although it still retained 90-fold selectivity to G12C, and enabled us to determine the crystal structure of its complex with HRAS(WT) at 1.98 Å resolution. Overlay of the two structures and molecular dynamics (MD) simulations revealed that there are no significant differences in the backbone conformation between HRAS(WT) and HRAS(G12C) captured by the respective monobodies (Supplementary Fig. 6a–c). Differences were restricted to the orientations of a few side chains in Switch II, an inherently flexible region[30]. Similar degrees of differences in side-chain orientations were also observed in other crystal structures of HRAS(WT), thus eliminating side-chain orientation as a main energetic contributor to the high selectivity of 12VC1 for the RAS mutants (Supplementary Fig. 6c)[30,31]. Taken together, these findings demonstrate that RAS mutations at position 12 can be selectively recognized in a noncovalent manner by a small pocket within a binding protein and potentially other types of molecules.

## 12VC1 Potently inhibits signaling mediated by endogenous RAS mutant.
The selectivity and potency of 12VC1 allowed us to perform proof-of-concept evaluation on whether a mutant-selective, noncovalent reagent can effectively inhibit the growth of cancer cells driven by endogenous KRAS mutants by using monobodies as genetically encoded, intracellular reagents[32,33]. We generated several stable cancer cell lines that express 12VC1 or MB(Neg), a non-binding monobody, fused to a fluorescent protein under the control of a doxycycline (dox)-inducible promotor (Supplementary Fig. 7a–c), and tested the effects of 12VC1 expression. To precisely examine the effect of monobody expression on cell proliferation, we set up a competitive growth assay. We mixed the dox-inducible cancer cell lines with their respective parental cell lines at one to one ratio, and monitored

the percentage of monobody-expressing cells, i.e., fluorescence-positive cells using flow cytometry. 12VC1 potently inhibited ERK activation and proliferation of cell lines harboring KRAS (G12C) (H358 and H23) and KRAS(G12V) (PATU8902 and H441), but not those with KRAS(G12D) (HPAF-II) and BRAF (V600E) (A375) or WT RAS (HEK293T) cell lines (Fig. 3a, b, Supplementary Fig. 8a, b). We then compared sustained expression of 12VC1 with sustained administration of the covalent inhibitor ARS1620 (Supplementary Fig. 9). Both 12VC1 and ARS1620 achieved similar levels of inhibition of RAS-mediated signaling and viability. These results show that a mutant-selective noncovalent inhibitor targeting the active state can be as effective as a covalent inhibitor, even though noncovalent inhibitors are subjected to reversible binding kinetics and thus under constant competition with RAS effectors[11]. Our results also suggest the importance of high selectivity in achieving efficient inhibition of mutant RAS, which is also underscored by the effectiveness of 12VC1 toward the KRAS(G12V) mutant in inhibiting RAS-mediated signaling (Fig. 3a and Supplementary Figs. 3a, 8a) despite having only moderate affinity to the active state of KRAS (G12V) ($K_D$ ~100 nM; Fig. 1b).

Next, we evaluated the anti-tumor activity of 12VC1 using a mouse xenograft model. We performed the xenograft experiment using PATU8902 harboring the KRAS(G12V) mutation, because targeting KRAS(G12V)-driven cancer is an unmet need whereas multiple G12C covalent inhibitors exist that are effective against KRAS(G12C) in xenograft experiments[6,9]. We prepared two separate derivatives of PATU8902, a cancer cell line of pancreatic origin, that express 12VC1 or MB(Neg) under the control of a dox-inducible promoter and injected them subcutaneously into nude mice. Expression of each monobody was induced after the average tumor size exceeded 100 mm$^3$ by replacing the regular mouse feeds with dox-containing feeds through the end of the xenograft experiment. 12VC1 expression significantly reduced tumor growth, whereas expression of MB(Neg) had no impact (Fig. 3c, Supplementary Fig. 8c). We noted that the MB(Neg) cell lines grew more slowly than the 12VC1 cell line. Although the two stable cell lines were generated from the same parental cell line, the integration of the retroviral vector can occur at different locations, which may have somehow made MB(Neg)-expressing cells grow slower. Clearly, such a variation across different cell lines is an inherent limitation of genetically encoded reagents. The exact cause of the large variability in the growth of the MB (Neg) group (Supplementary Fig. 8c) is unknown, but we suspect technical variability in tumor injection in this experiment. We did not observe such large variations in a similar experiment using the same retroviral vector system (see the next section).

At the end of the xenograft experiment, we no longer detected expression of 12VC1 from these tumors, although the cell used for injection robustly expressed 12VC1 when grown in dishes (Fig. 3d, top panels). By contrast, MB(Neg) was still readily detected from the control tumors as well as the equivalent cells grown in dishes (Fig. 3d, bottom panels). These results indicate that 12VC1-expressing cancer cells were eliminated and that tumor cells that did not express 12VC1 proliferated. These proliferating cells may have either lost 12VC1 expression due to silencing, or they did not express the monobody to begin with, which is a probable scenario, given that a small fraction of cells without monobody expression was present in the polyclonal population of the stable cell line (Supplementary Fig. 7c). This view is supported by the results from the in vitro proliferation assay (Fig. 3b, Supplementary Fig. 8b) showing that monobody-expressing KRAS(G12C) and KRAS (G12V) cells were rapidly depleted, whereas the monobody-expressing population was fairly constant over 8–9 days in non-RAS mutant cells. The similar pERK levels observed for tumors of the 12VC1-encoded cell line

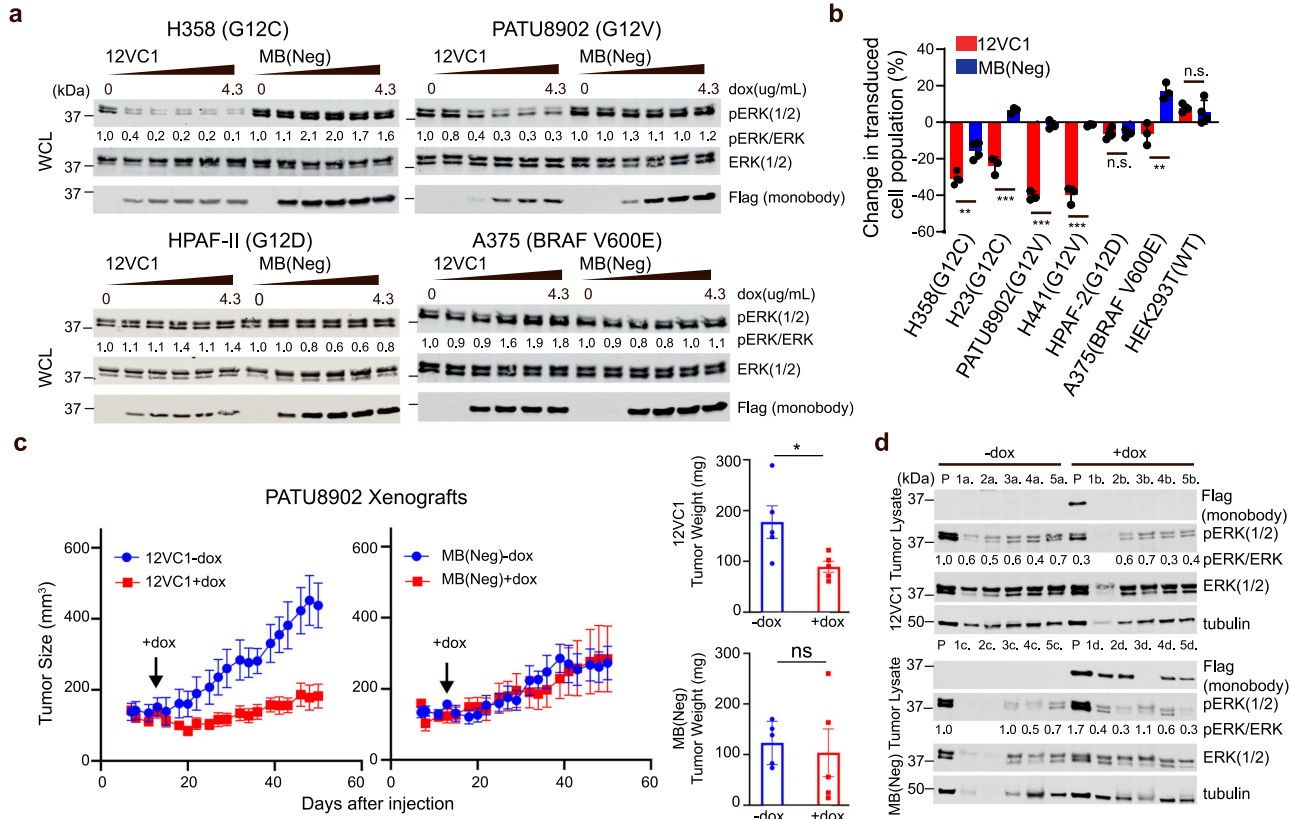

**Fig. 3 Inhibition by intracellularly expressed 12VC1 monobody of signaling and proliferation of RAS mutant-driven cancer cells. a** Effects of 12VC1 expression on ERK activation (24 h. induction with 4.3 μg/mL dox in quarter decrements). The numbers under the pERK panel indicate the ratio of pERK signal to the total ERK signal normalized to the no dox sample. Representative data shown, and experiment was reproduced at least one additional time with similar results. **b** Percent-change of monobody expressing population after 72 h of dox induction relative to 24 h of induction (mean ± SD). A negative percentage signifies a decrease in population. The *p*-values for the differences between 12VC1 and MB(Neg) expression, as determined with two-tailed unpaired *t*-test for H358, H23, PATU8902, H441, HPAF-II, A375, and HEK293T are 0.002, <0.001, <0.001, <0.001, 0.9, 0.006, and 0.55, respectively. Error bars represent s.d.; biological replicates, *n* = 4 for H358 and HEK293T and *n* = 3 for the other cell lines. **c** The effects of monobody expression on tumor growth in a mouse xenograft model. Tumors were developed from subcutaneously injected PATU8902 cells that express 12VC1 or MB(Neg) under a dox-inducible promoter. Mouse were given dox-containing feeds on the day indicated with the black arrow through the end of the experiment. Plots show the effect on average tumor sizes over time (*n* = 5, biological replicates, mean ± SEM). Extracted tumor weights at the end of the experiment are also shown (*n* = 5, biological replicates, mean ± SEM), average tumor sizes were compared using two-tailed unpaired *t*-test, *p*-value = 0.03, 0.71, respectively for 12VC1 and MB(Neg) tumors with and without doxycycline. **d** Immunoblotting for monobody expression and ERK activation in lysates from each tumor, numbered 1 through 5 at the end of the experiment. The lanes labeled P are the lysates of equivalent cells grown in plastic dishes confirming that these cells exhibited detectable pERK levels and dox-inducible expression of monobodies. The pERK levels of the tumor lysates were quantified relative to that of the cultured cells in the absence of dox (the leftmost lanes). Note that the cells grown in dishes and those in xenograft are distinct samples, and thus the pERK levels between these two types of samples may be substantially different. Immunoblotting has been technically replicated three times with similar results.

grown in the presence and absence of dox (Fig. 3d; normalized pERK intensities of 0.6 ± 0.1 and 0.5 ± 0.2 (mean ± SD), respectively) suggest that cells that remained at the end of the xenograft experiment in the dox-induced and -uninduced arms are similar in nature, further supporting our view. Together, these results demonstrate that mutant-selective inhibition by a noncovalent inhibitor can effectively suppress the growth of solid tumors in a mouse xenograft model.

**Selective degradation of RAS mutants using 12VC1 as a warhead.** We next tested whether 12VC1 can be used as a targeting ligand for constructing a PROTAC-like degrader selective to KRAS mutants. Monobodies fused to an E3 ubiquitin ligase subunit, VHL, have been developed that can degrade the protein of interest[34], including a VHL–NS1 monobody fusion that degrade RAS[27]. To minimize potential ubiquitination and subsequent degradation of VHL–monobody fusion proteins, we developed variants of 12VC1 termed 12VC1.1 and 12VC1.2 that

contained fewer Lys resides and fused them to VHL (Supplementary Table 1; Supplementary Fig 10a, b). Interestingly, 12VC1.2, which had the weaker affinity to KRAS(G12C) was the more efficient warhead at degrading KRAS(G12C) than 12VC1.1 (Supplementary Fig. 10c). In contrast, 12VC1.1, which has the stronger affinity of the two to KRAS(G12V), appeared to degrade RAS(G12V) more efficiently in PATU8902 (Fig. 4b, Supplementary Fig. 10d). These results suggest the presence of an optimal range of affinity (or the rate of dissociation that is generally correlated with affinity) for efficient RAS degradation and the utility of protein-based tools with readily tunable affinity in optimizing PROTAC design.

The VHL–12VC1.2 fusion was also more effective in reducing the pERK level than the higher affinity 12VC1.1 counterpart, even though the 12VC1.2 fusion was less abundantly expressed (Supplementary Fig. 10c). These results strongly suggest that the contribution of direct inhibition by the monobody warheads of these degraders to the reduction of RAS-mediated signaling was negligible under these conditions. They further emphasize

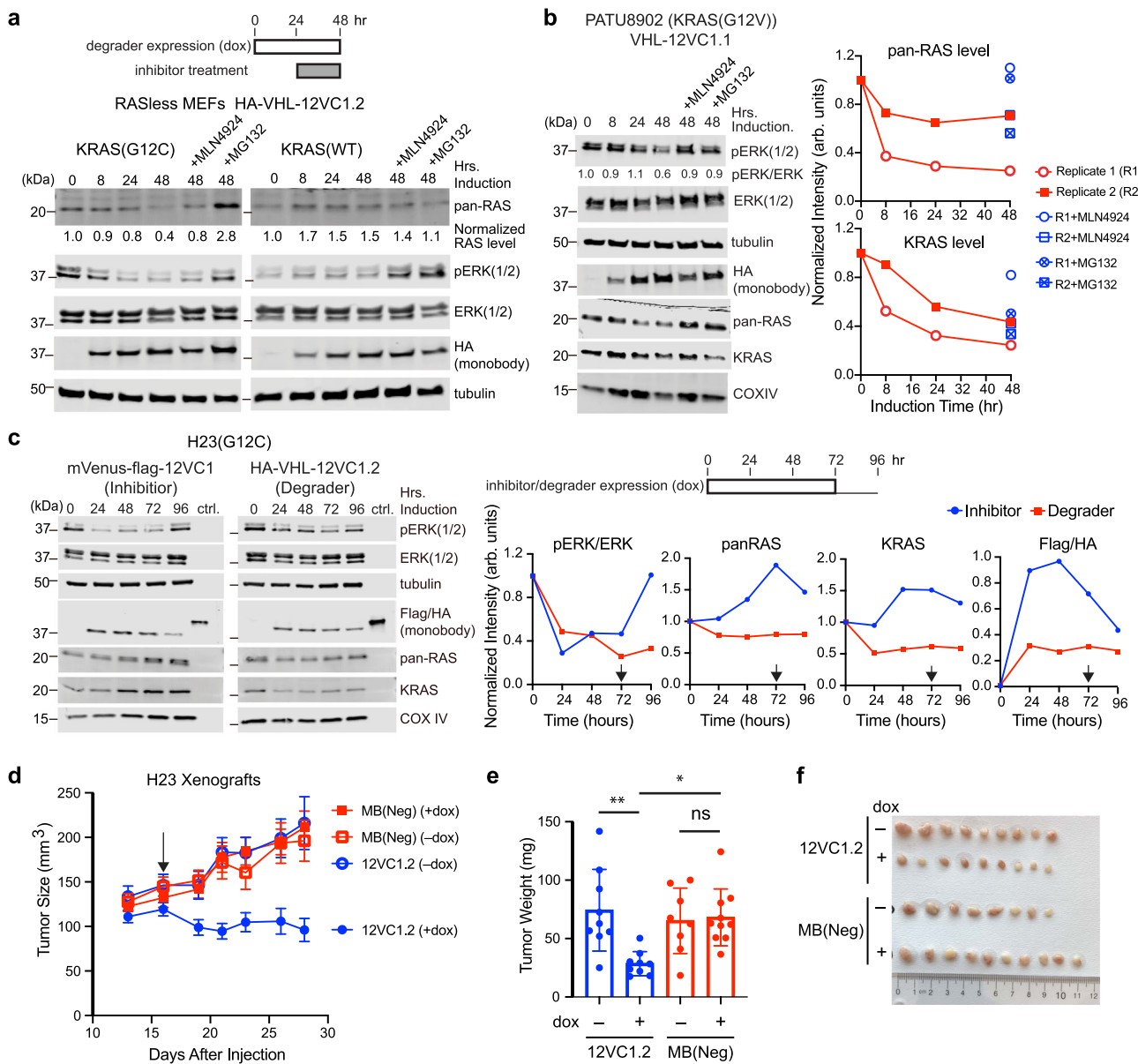

**Fig. 4 Selective degradation of RAS mutants with a VHL–monobody fusion protein. a** The time courses of the KRAS(G12C) and KRAS(WT) levels and the pERK level in RASless MEFs after the induction of the expression of the VHL–12VC1.2 degrader. MLN4924 (1 μM) or MG132 (5 μM) were added after 24 h of degrader expression for an additional 24 h (see the scheme). Total RAS was quantified and normalized relative to $t = 0$. **b** Degradation of endogenous KRAS(G12V) by a monobody degrader and its effect on the pERK level in PATU8902. The experimental schematic was identical to that for (**a**). The graphs show the quantification of the total RAS levels and KRAS levels from immunoblots (biological replicates, $n = 2$). **c** The time courses of the RAS levels and pERK level after the expression of a monobody inhibitor (blue) or a degrader (red) in the H23 cells (biological replicates, $n = 3$, representative results shown). A Flag-tagged mVenus–12VC1 fusion (inhibitor) and a HA-tagged VHL–12VC1.2 fusion (degrader) were expressed upon addition of doxycycline (1 μg/mL). The expression levels of the inhibitor or degrader were quantified using known amount of protein containing both Flag and HA tags (ctrl) as references. After 72 h of induction (black arrow), the media were replaced with serum- and doxycycline-free media to examine the persistent effects of intracellular inhibitors or degraders. **d** Mouse xenograft experiments with the H23 cell line expressing the VHL–monobody fusion proteins. Mice (5 per group for 12VC1.2 and MB(Neg)+dox, 4 per group for MB(Neg)−dox) were subcutaneously injected at the right and left flanks with H23 cell lines that express either fusion protein under a dox-inducible promotor, with the exception of one mouse in the 12VC1 group that was injected only at the right flank with tumor cells. One group of mice per cell line was given dox-containing feeds when the average tumor size was above 100 mm³ (black arrow) through the end of the experiment. The sizes of the tumors in MB(Neg) group (red squares) and 12VC1.2 group (blue circles) with (solid symbols) and without (empty symbols) dox feeds are plotted (mean ± SEM) as a function of time after cell injection. **e** Tumors were extracted at the end of experiment and their weights were compared ($n = 8$ and 10 for tumors for MB(Neg) without and with dox, respectively, and $n = 9$ tumors for 12VC1.2, two-way ANOVA, 12VC1.2 −dox vs +dox, $p$-value = 0.004, 12VC1.2 +dox vs MB(Neg) +dox, $p$-value = 0.01, MB(Neg) +dox vs −dox, $p$-value > 0.99). The bar centers show the average tumor weights of the population and the error bars show SD. **f** An image of extracted tumors from each group with a ruler for scale, ranked from large to small in weight.

that a potent inhibitor with high affinity may not be an ideal warhead for constructing an efficacious PROTAC.

These VHL–monobody fusions selectively and efficiently degraded KRAS(G12C) in RASless MEFs (Fig. 4a, Supplementary Fig 10c, e), as well as endogenous RAS in PATU8902 and H23 cancer cell lines (Fig. 4b, c). We observed a decrease of roughly 30% of total RAS and 50% of KRAS, which is consistent with estimations of a RAS mutant populating 15–40% of total RAS from cell lines containing heterozygous RAS mutation[35]. Treating the cells with MLN4924, a neddylation inhibitor, or MG132, a proteasome inhibitor, rescued the RASless MEFs and PATU8902 from RAS degradation (Fig. 4a, b, Supplementary Fig. 10e), supporting the mechanism of proteasome-dependent degradation. These inhibitors did not increase expression levels of VHL–monobody fusions (Fig. 4a, b, Supplementary Fig. 10e), which strongly suggest that the VHL–monobody fusions underwent minimal degradation and that these constructs were able to engage and degrade multiple RAS mutant molecules during their life-cycle, a key attribute for achieving effective target degradation.

The development of a selective degrader for a KRAS mutant made it possible to compare effects of mutant-selective inhibition and mutant-selective degradation in a signaling assay (Fig. 4c). Treatment with the covalent inhibitor, ARS1620, or our noncovalent 12VC1 inhibitor resulted in an increase in the endogenous RAS level within 48 h after the initiation of the treatment (Fig. 4c, Supplementary Fig. 9b). Similar increases of endogenous RAS after prolonged treatment with covalent inhibitors have been reported[6,36], and increased synthesis of RAS has been proposed as a mechanism of adaptation against covalent inhibitors[6,37]. In contrast, the degrader kept the KRAS and pan-RAS levels reduced throughout the experimental period (Fig. 4c). Both inhibitor and degrader substantially suppressed the pERK level as their expression was induced. However, the pERK level in the inhibitor-treated cells rebounded even while the inhibitor level was high and further increased after the withdrawal of dox, whereas the pERK level of degrader-treated cells continued to decline (Fig. 4c). The ability of the degrader to suppress the level of the KRAS mutant, rather than allowing it to rapidly rebound, probably contributes to its ability to extend the inhibition of RAS-mediated signaling. The slow decay of this particular degrader may also positively contribute. Thus, these results highlight the potential advantage of a noncovalent degrader over a noncovalent inhibitor against KRAS mutants.

Finally, we examined the effectiveness of mutant-selective degradation in a mouse xenograft model using the H23 cell line (Fig. 4d). We chose the H23 cell line harboring KRAS(G12C) for this xenograft experiment, because it does not respond well to direct inhibition with a G12C-specific covalent inhibitor[9]. It also generated a high percentage of monobody-expressing cells without sorting, which minimized complications caused by non-expressing cells as we discussed above. Tumors expressing the VHL–12VC1.2 fusion were significantly smaller than non-induced tumors and tumors expressing the VHL-MB(Neg) fusion (Fig. 4d–f). Recovered tumor cells contained a reduced amount of VHL–12VC1.2 compared with that in the starting cells prior to injection and that of the negative control degrader in the recovered tumor cells (Supplementary Fig. 11). These results suggest that most of the cells expressing VHL–12VC1.2 fusions did not proliferate. In light of a recent report demonstrating that a covalent PROTAC was able to target and degrade KRAS (G12C)[25], our work further demonstrate that selective degradation of endogenous RAS mutants can effectively reduce the proliferation of RAS-driven tumors and that non-covalent and selective degradation of the endogenous RAS mutants is feasible.

## Discussion

Using monobodies as "tool biologics", this work has substantiated that oncogenic RAS mutants can be selectively recognized and inhibited using noncovalent approaches. Monobody 12VC1 bound strongly to the active state of RAS mutants and not to the active or inactive states of wild-type RAS and minimally to other intracellular proteins (Fig. 1). Since the development of the first covalent inhibitor against RAS mutant G12C, there have yet to be breakthroughs in the development of inhibitors selective to other RAS mutants that show efficacy in both cell-based studies and animal models. The development of 12VC1 has addressed a major question in RAS drug discovery in the recent years, that is, the feasibility of developing an inhibitor that selectively recognize the mutant in a noncovalent manner. Selective targeting of RAS mutants is likely to be crucial for the development of future therapeutics against RAS-driven cancers that are well tolerated, because wild-type RAS is ubiquitously expressed in all tissues and plays important roles including maintaining homeostasis and immune responses. 12VC1 showed specificity toward multiple RAS mutants (Fig. 1b, Supplementary Fig. 1b). This broad spectrum may be therapeutically important in light of reports describing the presence of multiple RAS mutations in a single patient[38].

The structure of the monobody-RAS interface will guide the design of noncovalent inhibitors (Fig. 2). We demonstrated that a single inhibitor that possesses an appropriate pocket in the binding interface is capable of selectively recognizing multiple oncogenic RAS mutants. Although it is difficult to recapitulate this shallow and rigid pocket architecture using free-standing small molecules, this challenge may be overcome by designing larger molecules, such as peptidomimetics and macrocycles using the 12CV1 structure as the starting point[16,39–41].

12VC1 showed potent efficacy in inhibiting the signaling and proliferation of RAS-driven cancer cell lines containing KRAS (G12V) and (G12C) and in a mouse xenograft model (Fig. 3). Although impactful in establishing the effectiveness of mutant-selective, noncovalent inhibition, these experiments also highlighted limitations of genetically encoded reagents. The xenograft experiments utilized polyclonal populations of retrovirally transduced cells that had diverse levels of monobody expression. By the end of the experiment, tumor cells expressing no or little monobody dominated the surviving tumors, which made it difficult to accurately determine the impact of the monobody on tumor growth and signaling in a long-term experiment. Recent reports have demonstrated the possibility of delivering functional biologics across the cell membrane[42,43], which suggests the feasibility of treating all cells within a tumor with 12VC1 or other monobodies. When combined with such technologies, 12VC1 itself has substantial potential to aid the future drug development effort against RAS mutants. Alternatively, nucleic acids encoding it can be delivered using gene delivery technologies including those that are being extensively tested in vaccine development against SARS-CoV-2[44,45].

Our results conclusively established that endogenous RAS mutants can be selectively degraded in a noncovalent manner (Fig. 4). Expression of 12VC1-based degraders significantly reduced tumor size in mice. It will be beneficial to directly compare the effectiveness of a degrader and inhibitor made with an identical noncovalent warhead in inhibiting tumors in vivo. We also showed that both noncovalent and covalent inhibition of a RAS mutant led to an eventual increase in production of the RAS mutant in cells (Supplementary Fig. 9b). Selective degradation of RAS combats this feedback (Fig. 4c), which strongly support the potential of mutant-selective RAS degraders as an effective therapeutic strategy.

VHL–12VC1 fusions represent the second examples of degraders that utilize RAS-targeting monobodies as the target engagers[27,46]. A recent report using the NS1 monobody as a warhead for a degrader (NS1 was referred to as aHRAS) has demonstrated that HRAS was preferentially degraded over KRAS[27], despite the fact that NS1 binds to both HRAS and KRAS[20]. VHL–NS1 was no more effective than NS1 alone in inhibiting cell growth, although the fusion does not appear to be rapidly degraded. In contrast, VHL–12VC1 was efficacious in selectively degrading KRAS mutants and more potent at inhibiting RAS-mediated signaling and proliferation than 12VC1 alone. The low efficacy of VHL–NS1 against KRAS is likely due to its lower affinity to KRAS than to HRAS so that HRAS served as a target sink, although we cannot exclude the possibility that the configuration of VHL–NS1 is not as effective in RAS ubiquitination as that of VHL–12VC1. This comparison strongly suggests the importance of high selectivity of the target engaging moiety in PROTAC-like molecules. The recent success of degrading and inhibiting KRAS mutants using degraders utilizing a KRAS-selective DARPin[26,46], which should not be inhibited by the other RAS isotypes, supports this view.

We demonstrated that the affinity of the warhead to the target is important for developing effective degraders by tuning the affinity of the monobody warhead by mutations. The VHL–12VC1 fusion experienced minimal degradation even in the absence of its target (Fig. 4a). This is in contrast to DARPin based PROTAC where significant reduction of VHL-fused DARPin was observed[26]. The DARPin scaffold contains 9 surface-exposed lysine residues, whereas 12VC1.2 and 12VC1.1 contains 2 and 3 lysine residues, respectively. This comparison suggests the success of our design that minimized Lys residues in the monobody unit. Although not examined in this work, it is equally straightforward to produce many variants of the linker between VHL and monobody units and also replace VHL with other E3 ligase subunits[26]. These design flexibilities, in addition to the ability to achieve exquisite selectivity, may further increase the utility of monobodies as tool biologics for rapidly examining the effectiveness of the direct inhibition and degrader approaches against novel targets.

The ability of the degrader to engage multiple targets within its lifetime reduces the effective concentration required for inhibition compared with occupancy-based inhibition. This feature suggests an approach to enhance the potency of intracellular biologics, thus to reduce the required level of intracellular delivery, and consequently to facilitate the development of "cell-penetrating" biologics targeting intracellular proteins.

## Methods

**Protein expression and purification.** All proteins used for monobody development and binding assays, including RAS constructs (KRAS4B residues 1–174 containing G12C, G12V, G12D, G13D, WT, NRAS(WT), and HRAS(WT)), monobodies, and RAF-1 RBD (residues 51–131) were produced with an N-terminal tag containing His6, Avi-tag for biotinylation and a TEV protease recognition site using the pHBT vector[19]. The RAS constructs were made by Kunkel mutagenesis using the KRas_G12x_rev set of primers and primer KRas_addKEKMSKDG_rev (Supplementary Table 4). Monobody genes were cloned into the pHBT vector by sticky-end PCR using the FN5 and FN3 set of primers. The DNA sequences were confirmed using a primer that anneals to the T7 promoter region (Supplementary Table 4). The proteins were produced in *E. coli* BL21 (DE3). To produce biotinylated proteins, *E. coli* BL21(DE3) with the pBirA plasmid was used as the host and grown in the presence of 50 μM of biotin. Expressed proteins were purified using Ni-Sepharose columns (GE Healthcare) via gravitational flow, followed by dialysis in Tris-buffered saline (TBS, 50 mM Tris-Cl pH 7.5 containing 150 mM NaCl) for non-RAS proteins. TBS containing 20 mM MgCl2 and 0.5 mM DTT was used for RAS proteins. Samples were further purified using a Superdex 75 size exclusion column on an AKTA Pure systems (GE Healthcare). HRAS(WT) and HRAS(G12C) (residues 1–166) used for crystallization was produced as a fusion protein with His6 and yeast SUMO at the N terminus using an in-house vector described previously[47]. The tag was removed with SUMO hydrolase, followed by overnight dialysis in RAS buffer (50 mM Tris-Cl pH 7.5,

150 mM NaCl, 20 mM MgCl2, 5 mM BME). For monobodies (12VC1 and 12VC3) used in crystallization and RAS used in BLI experiments, proteins were cleaved with TEV-protease in the presence of 0.5 mM EDTA and 1 mM DTT after the Nickel column purification step. Cleaved tags and His-tagged ySUMO hydrolase or TEV-proteases were removed by passing samples through a Ni-Sepharose column, followed by size-exclusion chromatography as described above.

**Nucleotide exchange of RAS.** Purified RAS proteins used in binding experiment and crystallization were prepared by diluting stock protein (typically containing 20–250 μM RAS) 25 times with 20 mM Tris-Cl buffer pH 7.5 containing 5 mM EDTA, 0.1 mM DTT, and 1 mM final concentration of a nucleotide (GDP or GTPγS). Samples were incubated at 30 °C for 30 min. MgCl2 was then added to the sample at a final concentration of 20 mM and the solution was further incubated on ice for at least 5 minutes prior to use.

**Monobody development.** General procedures for the development of monobodies against purified protein targets have been described previously[18,19,48,49]. After four rounds of phage display library selection using biotinylated KRAS(G12C) at concentrations of 100, 100, 50, and 50 nM, the genes encoding monobodies from the enriched phage pool were transferred to a yeast display vector, which was used to construct yeast-display libraries. The yeast display libraries were sorted using a Bio-Rad S3e fluorescence-activated cell sorter or a FACSARIA II cell sorter (BD Biosciences). The first round of sorting recovered clones that bound to KRAS (G12C); the second round recovered clones that did not bind to KRAS(WT); and the third round recovered clones that bound to KRAS(G12C). Single clones were then screened for selective binding to RAS mutants. The expression of monobodies on the surface of yeast cells were detected using mouse anti-V5 (ThermoFisher, MA5-15253, 1:75 for sorting approximately 10^7 yeast cells, and 1:300 for staining 10^5 yeast cells for analysis) follow by labeling using anti-mouse IgG-FITC conjugate (Millipore Sigma, F0257, 1:100). Target binding was detected with neutravidin Dylight650. Yeast cells were analyzed using an iQue flow cytometer (Sartorius) (Supplementary Information Fig. 1b). The median of the signal intensity in the Dylight650 channel for the 75–95th percentile population was taken as representative signal. This sampling method of flow cytometry events minimizes erroneous contributions from events with anomalously high signals while retaining events with high signals. For degradation experiment, monobody degraders were developed by fusing Monobodies 12VC1.1 and 12VC1.2 C-terminal to the VHL domain (1-213) with a SSSSG linker and N-terminal HA tag (YPYDVPDYA) following a published design[34].

**Biolayer interferometry (BLI) analysis.** BLI experiments were performed on an Octet Red96 instrument (Molecular Devices). Biotinylated monobodies were immobilized on streptavidin biosensor tips. Samples were diluted in 20 mM HEPES-NaOH buffer (pH 7.4) containing 150 mM NaCl, 5 mM MgCl2, 0.2 mM TCEP and 0.005% Tween-20. BLI signals were analyzed using Octet Data Analysis software (Molecular Devices).

**Cell culture.** All cell lines used in the study were either directly purchased from ATCC (HEK293T, A375, HPAF-II) or validated externally via IDEXX (PATU8902, H23, H358). HEK293T, Flp293, PATU8902, A375, and HPAF-II cells were maintained in DMEM high glucose with L-glutamine (Hyclone) supplemented with 10% FBS (Gemini Bio-products) and antibiotics-antimycotics (Gibco). H358, H23, and H441 cells were maintained in RPMI-1640 high glucose with sodium pyruvate, L-glutamine (Thermo), supplemented with 10% FBS and antibiotic-antimycotic. The absence of mycoplasma contamination was periodically confirmed using a PCR-based mycoplasma testing kit (LiLIF).

**Transient expression of KRAS and monobody for confocal imaging and signaling experiments.** HEK293T cells were cultured in glass-bottom 8-well chambers (ibidi GmbH) for colocalization assays or in a 12-well plate for signaling assays for 1 day prior to transfection. On the day of transfection at 70–90% confluency, media were replaced with antibiotics-free complete media (DMEM supplemented with 10% FBS). The vectors encoding mCherry-fused monobodies were constructed by cloning their genes amplified using primers, NheI_mcherrF mcherr_-flagR, Flag_Mb_F, and mb_ApAI_R (Supplementary Table 4), into the pEGFP vector at the NheI and ApAI restriction enzyme sites. The pEGFP KRAS4B vector was a gift of Prof. Mark Philips. Transfection of pEGFP vectors encoding the appropriate mCherry fused monobodies and EGFP fused KRAS4B constructs was performed with lipofectamine 3000 (Thermo Fisher Scientific) and according to the manufacturer's recommended protocol. On the following day, transfected cells were imaged with a LSM710 confocal microscope (Zeiss) for colocalization experiment or harvested for western blot analysis.

**Pull-down assays.** H358, PATU8902, and HEK293T cells were cultured in 10 cm plates (Corning, #430167). H358 and PATU8902 cells were treated with and without ARS1620 (SelleckChem, S8707) at a final concentration of 10 μM for 1.5 h, and HEK293T cells were treated with EGF (PeproTech; AF-100-15) at a final concentration of 50 ng/mL for 4 min. Cells were lysed by incubating them on ice

for 15 min in GTPase lysis buffer (25 mM Tris-Cl pH 7.2, 150 mM NaCl, 5 mM MgCl₂, 1% NP-40, and 5% glycerol supplemented with protease tablet (Roche, 5892991001) and phosphatase inhibitors (1 mM sodium orthovanadate, 10 mM NAF, 54 mM β-glycerol phosphate)) immediately before analysis. After centrifugation for 15 min at 15,000 × g, the supernatants were collected and incubated with SA agarose resins (Thermo Fisher Scientific) for 1 h at 4 °C to remove non-specific binders to the resins. After the removal of resins via centrifugation, the pre-cleared lysates were then incubated with biotinylated monobodies bound to SA agarose resins for 3 h at 4 °C while rotating. The agarose resins were then washed twice with the GTPase lysis buffer and boiled in 1x SDS buffer with 37.5 mM β-mercaptoethanol and processed for western blotting. RAS proteins were detected using a mouse pan-RAS antibody (SCBT, sc-166691, 1:500) followed by anti-mouse HRP conjugate (Pierce, 31432, 1:4000) with the ECL 2 Western blotting substrate (Pierce; 80196).

**Cell signaling assay and quantification of immunoblots.** Cells were lysed in RIPA buffer (50 mM Tris-Cl pH 8, 150 mM NaCl, 5 mM EDTA pH 8, 0.1% SDS, 1% NP40) supplemented with protease inhibitors (Roche; 5892991001) and phosphatase inhibitors (54 mM β-glycerophosphate, 10 mM NaF, 1 mM sodium vanadate) for 15 min on ice, and centrifuged at 15,000 × g for 15 min. The supernatant was collected and measured for total protein amount using BCA assay (Thermo Scientific; 23227). Lysate (7–20 µg per well) was loaded onto a pre-cast SDS gel (BioRad; 456-1096) followed by electrophoresis. The proteins were then transferred from the gel to a low fluorescence background PVDF membrane (Millipore; IPFL00010) or regular PVDF membrane (Millipore; ISEQ00010) depending on whether fluorescence or chemiluminescence detection was performed. Phospho-ERK was detected using a rabbit anti-pERK antibody (Cell Signaling, 9101S, 1:1000) and total ERK was detected using a rabbit anti-ERK antibody (Cell Signaling, 9102S, 1:1000). Monobody expression was detected using mouse anti-Flag antibody (Sigma, F3165, 1:2000) or anti-HA tag antibody (Bio-Legend, 901516, 1:2000). Loading controls were detected with primary antibody against tubulin-alpha (Thermo Scientific, 62204, 1:5000) or against cytochrome oxidase (COX IV) (Li-Cor, 926-42214, 1:2000). The total RAS level was detected using a pan-RAS antibody (SCBT, sc-166691, 1:500 for endogenous RAS and 1;2000 for overexpressed RAS) or a pan-RAS antibody of rabbit origin at 1:500 dilution (CST, 3965S, 1:500) for lysates prepared from tumors in mouse xenograft experiments. The KRAS level was determined using a KRAS-specific antibody at 1:500 dilution (Sigma; WH0003845M1). For fluorescence detection of immunoblots, the membranes were imaged with a Licor Odyssey Clx imager (Li-Cor Bioscience) using IRDye anti-Rabbit 800CW and anti-Mouse 680LT (Li-Cor Bioscience, 926-32211 and 926-68020, respectively, 1:5000 for both) as secondary antibodies. For chemiluminescence detection, the membrane was imaged with a ChemiDoc imager (BioRad) using anti-Mouse or anti-Rabbit secondary antibody conjugated to HRP (Pierce; 31432 and 31462, respectively, 1:4000 for both). Band intensities of western blot were analyzed with Image Studio Lite version 5.2 (Li-Cor Bioscience). To quantify protein abundance in the degradation experiment, the intensities of pERK, RAS, KRAS, and Flag/HA tag bands were first normalized against the loading control of the perspective blot, which were either tubulin-α, COX IV or total ERK, followed by subsequent normalization to the zero time point. The relative concentrations of intracellular FLAG-tagged monobody inhibitor or HA-tagged monobodies were determined by comparing the intensity of protein bands with the intensity of the bands from equal amounts of loading control proteins (GenScript, M0101) containing both FLAG and HA tag.

**Proteomics analysis.** Protein samples captured by monobody immobilized on Dynabead M-280 streptavidin (Thermo Fisher Scientific) were digested on beads using trypsin. An aliquot was loaded onto an Acclaim PepMap trap column (2 cm) in line with an EASY-Spray analytical column (50 cm) using the auto sampler with either a data-dependent mode on an Easy-nLC 1000 interfaced to a Thermo Fisher Scientific Q Exactive mass spectrometer or a targeted analysis on an Easy-nLC 1200 interfaced to a Thermo Fisher Scientific Q Exactive HF-X mass spectrometer for peptides specific to either the KRAS(G12V) mutant or KRAS/KRAS 2B. All Acquired MS2 spectra were searched against the Uniprot *homo sapiens* reference database containing common contaminant proteins and the mutant KRAS4B (G12V) using Sequest within Proteome Discoverer 1.4 for the data-dependent analysis and using Byonic for the targeted analysis search. The significance analysis of interactome (SAINT) scores and enrichment scores were calculated using spectral counts of the captured proteins as described previously[50]. This method adds a value of 0.1 to the spectral counts to avoid divisions by zero when calculating enrichment. The spectral counts were normalized to the length of the protein and the total number of the spectra from the affinity purified sample. Further details are provided in Supplementary Method 1.

**Crystallization and X-ray structure determination.** Purified and tag-cleaved H-RAS(G12C) or H-RAS(WT) bound to GTPγS were incubated with purified monobody 12VC1 or 12VC3 at a 1:1.1 molar ratio. The complexes were purified with a Superdex 75 10/300 SEC column (GE Healthcare) in 20 mM Tris-Cl buffer pH 8 containing 100 mM NaCl, 20 mM MgCl₂, and 0.2 mM TCEP and concentrated to approximately 10 mg/mL. Both complexes (12CS1:HRAS(G12C) and

12VC3:H-RAS(WT)) were crystallized in 0.225 M sodium tartrate and 20% PEG3350 when mixed 1:1 in a total volume of 200 nl dispensed by a Mosquito crystallization robot (TTP Labtech) using the hanging drop vapor diffusion method. Crystals were preserved in the mother liquid plus additional 10% (v/v) of PEG3350 for the HRAS(G12C):12VC1 complex and mother liquid plus 20% glucose for the H-RAS(WT):12VC3 complex. X-ray diffraction data were collected at the Advance Photon Source at the Argonne National Laboratory using beam line 19ID. Diffraction data were processed using HKL3000[51] and the starting model was built by molecular replacement using PDB entry 4g0n as search models using Phaser[52,53]. Refinement was performed by phenix refinement[54], Coot[55], and PDBredo[56].

**Computational analysis of crystal structures.** Fragment-centric topographical mapping (FCTM), as well as molecular dynamics (MD) simulations, were performed to investigate the underlying reasons for selective binding of monobody 12VC1 against different RAS mutants. FCTM was performed using *AlphaSpace*[28], which utilizes a geometric model based on Voronoi tessellation. Briefly, *AlphaSpace* identifies and represents all concave interaction space across the protein-protein interface as a set of alpha-atom/alpha-space pairs, which are then clustered into discrete fragment-centric pockets.

MD simulations were performed with the Amber14 molecular dynamics package, employing the Amber14SB force field for proteins[57]. The initial structures for each simulation system were constructed based on the crystal structure of the HRAS–12VC1 complex. The protonation states of charged residues at pH 7 were determined based on pKa calculations via the PDB2PQR server[58]. Each system was neutralized with Na⁺ counterions and solvated with explicit TIP3P water in a rectangular periodic box with 12.0 Å buffer. The Particle-Mesh Ewald (PME) method with 12.0 Å cutoff for the non-bonded interactions was used in the energy minimizations and MD simulations. After a series of minimizations and equilibrations[59], standard molecular dynamics simulations were performed on GPUs using PMEMD[60,61]. For each system, MD simulation was carried out for 250 ns with the periodic boundary condition and snapshots were saved every 10 ps. The SHAKE algorithm was applied to constrain all bonds involving hydrogen atoms and the Berendsen thermostat method were used to control the system temperature at 300 K. All other parameters were default values. Saved snapshots were analyzed using the *cpptraj* module in AmberTools 15.

Based on the crystal structure of the HRAS(G12C)–12VC1 complex, we carried out a series of molecular dynamics (MD) simulations to investigate how 12VC1 interacts with different single mutants of HRAS (G12C, G12V, G12D, G13D, and Q61L) as well as the wild type. The stability of the complexes was measured using the following metrics: (1) the average distance between residue 12 of HRAS and V33, A48, and K50 of monobody; (2) the average RMSD of monobody during MD simulation, with the RAS-monobody complex aligned based on RAS; (3) the average RMSD of G44, A45, and F46 of monobody during MD simulation, with the complex aligned based on RAS; (4) the average distance between K117 of RAS and G44 of monobody.

In addition, in order to investigate the stability of HRAS conformations captured by 12VC1 and 12VC3 monobodies, MD simulations of HRAS structures in the absence of monobody binding were carried out. The resulted MD snapshots were clustered and representative snapshots were compared with the two HRAS conformations in the monobody complex crystal structures and other 36 GTP analogue-bound HRAS crystal structures obtained from the PDB database (Supplementary Table 5)[62]. These MD simulations were carried out with the Amber 16 package[63] using the Amber FF14SB force field[57]. Each molecular system was neutralized by adding counter-ions, and was solvated in an explicit TIP3P water box. The GSP molecule was parametrized with general amber force field second version (GAFF2)[64]. AmberTools were used to prepare structures and analyze MD trajectories[65]. The DBSCAN method was used for MD snapshots clustering[66].

**Stable cell line generation.** Genes encoding EGFP and mVenus and genes encoding monobodies were cloned into the pRetro-TetOne vector (CloneTech) with a flag tag spacer using In-Fusion cloning (Takara Bioscience) using PCR fragments generated from the primer sets, Infusion_EcoRI_FP_F, mcher_flagR, Flag_Mb_F, and Infus_MB_BAM_R (Supplementary Table 4). The mVenus gene was a gift from Steve Vogel (Addgene Plasmid #27793)[67]. For degradation studies, genes encoding N-terminally HA-tagged VHL–monobody fusion proteins were cloned into the pRetro-TetOne vector using In-Fusion cloning using PCR fragments amplified using the primer sets, Infus_EcoRI_HA_VHL_F, VHL_SSSSG_R, SSSSG_MB_F, and Infus_MB_BAM_R (Supplementary Table 4). The VHL gene was a gift from William Kaelin (Addgene Plasmid #19999)[68]. The constructs in pRetroTetOne vectors were confirmed by sequencing using the TetOne_ForSeq and TetOne_RevSeq primers (Supplementary Table 4). Retroviruses were generated by co-transfecting the packaging cell line GP2-293 using Lipofectamine 3000 with the pRetro vector derivatives and the virus envelope vector pVSV-G. After 6 h, the transfection mixture was replaced with fresh complete media. Retroviral supernatant was collected 48 h post transfection and filtered using a 0.45 µm filter. Prior to retroviral transduction, cells of interest were plated in a 6-well cell culture plate (Thermo Scientific; 130184). Polybrene (final concentration of 4 µg/mL) was added to each well containing cells follow by viral transduction using 500 µL of

filtered viral supernatant. After the addition of retroviruses, the 6-well plate was centrifuged at $1200 \times g$ for 1.5 h to increase the transduction efficiency. After eight hours, viral supernatants were replaced with new complete media with Tet-Approved FBS (Takara Biosciences; 631367 or Gemini Biosciences, 100-800). Cells were sub-cultured in complete media containing 1 µg/mL puromycin for selection 48 h post transduction. After puromycin resistance was established, cell lines expressing VHL–monobody fusions were aliquoted and used for experiment at the lowest passage number possible. Cells expressing a fluorescent protein-fused monobody were sorted using FACS (Supplementary Fig. 1a). Prior to sorting, expression of the monobody was induced by adding 0.1 µg/mL of doxycycline (Clonetech; 631311) for 6 h. The short induction period and low doxycycline concentration were designed to alleviate potential stress caused by inhibition of RAS function by monobodies. Sorted cells were expanded and monobody expression upon induction was confirmed before they were used for experiments. Cells that expressed VHL fused monobody were not sorted due to a lack of selection marker. Cells were maintained in media supplemented with Tet-Approved FBS.

For generating stable cell lines that expressed EGFP fused KRAS, the Flp293 cells (Thermo Scientific) were cultured in complete DMEM supplemented with Zeocin (100 µg/mL) for at least 1 passage prior to co-transfection with pOG44 and pFRT vector derivatives (Thermo Scientific) contained cDNA encoding an EGFP fused KRAS construct of interest and hygromycin B resistance gene. 48 h after transfection, cells were expanded into media containing Hygromycin B (200 µg/mL) for selection. Individual colonies were then screened or pooled and sorted. The resulting cells were verified via flow cytometry for tightly distributed levels of EGFP expression.

**Cell proliferation assay.** Dox-inducible stable cell lines expressing 12VC1 or MB (Neg) were mixed with the parental cell line at an approximately 1:1 ratio and seeded in a 12 well or 24 well plate. Cells were then cultured in the presence of 1 µg/mL doxycycline for the entire duration of experiment (7–8 days) and subcultured into a new 12 well or 24 well plate every 2 days. For a control experiment to eliminate the possibility that the growth bias is caused by retroviral transduction but not by monobody expression, cells transduced with the 12VC1 vector were grown in the absence of dox and then induced 24 h prior to the last measurement. The mixed culture was sampled periodically and the ratio between monobody expressing cells (mVenus and GFP positive) and non-expressing cells was determined from measurements using a Gauva EasyCyt Flow Cytometer (Millipore Sigma) and analyzed with FlowJo (Supplementary Figs. 7, 12).

**Cell viability assay.** A total of 3000 cells were seeded in the well of a 96-well flat-bottom plate with a clear window. Cells were either treated with 15 µM of ARS1620 or 1 µg/mL doxycycline. Fresh reagents were added every 3 days. To measure cell viability the presto blue reagent (Thermo Fisher) was added at 10% of the culture volume. After 2 h of incubation at 37 °C, the fluorescence intensity of the wells was measured (ex:560 nm/em:590 nm) with a FlexStation 3 multi-mode plate reader (Molecular Devices).

**Mouse xenograft experiments.** Animal experiments were approved by NYU Langone Institutional Animal Care and Use Committee (IACUC, protocol 170602). PATU8902 or H23 cells expressing a monobody or a VHL–monobody fusion under a dox-inducible promotor ($5 \times 10^6$) were subcutaneously injected into female athymic nude mice 8-10 weeks of age (Strain CR ATH HO(490), Charles River). The mice were housed in an animal facility with a 12-h day and night cycle. Once the average tumor size exceeded 100 mm³, the mice were given food containing doxycycline (Envigo; TD.01306) through the end of the experiment. Tumor sizes were measured using a digital caliper thrice a week along with body weight to ensure that mice were healthy through the duration of the experiment.

**Ethical compliance.** This study was performed in compliance with all relevant ethical regulations. All animal experiments in this study were approved by the New York University Grossman School of Medicine Institutional Animal Care & Use Committee (IACUC).

**Reporting summary.** Further information on research design is available in the Nature Research Reporting Summary linked to this article.

## Data availability

Atomic coordinates for the HRAS(G12C)-12VC1 and HRAS(WT)-12VC3 structures have been deposited at the Protein Data Bank with the accession codes of 7L0G and 7L0F, respectively. The mass spectrometry raw files are accessible under MassIVE ID: MSV000086569. The data sets generated during and/or analyzed during the current study are available from the corresponding author on reasonable request. Source data are provided with this paper.

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

## Acknowledgements

The authors would like to acknowledge Beatrix Ueberheide and Jackeline Ponce of the Proteomics Laboratory at NYU Grossman School of Medicine for their technical assistance and helpful discussion; Michael Cammer of the Microscopy Laboratory  and the staff of the Cytometry and Cell Sorting Laboratory at NYU Grossman School of Medicine for assistance: Ankit Gupta and Kohei Kurosawa for assistance in cellular studies; Angel Sing for assistance with the mouse experiments; Mark Philips for the pEGFP-KRAS(WT) vector. The X-ray crystallography data were acquired at Beamline 19ID of the Advanced Photon Source (APS) at Argonne National Lab with the support of beamline scientists Jurek Opsipiuk and Kemin Tan. APS is a U.S. Department of Energy (DOE) Office of Science User Facility operated for the DOE Office of Science by Argonne National Laboratory under Contract No. DE-AC02-06CH11357. This work was supported by the National Institutes of Health grants R35 GM127040 (Y.Z.), R01 CA194864 (S.K.), R21 CA201717 (J.P.O.), and R01 CA212608 (J.P.O. and S.K.). K.W.T. was supported in part by NIH fellowship (F32 CA225131) and ACS fellowship (PF-18-180-01-TBE). The core facilities of NYU Grossman School of Medicine were partially supported by the Cancer Center Support Grant P30CA016087.

## Author contributions

K.W.T., A.K., Y.Z., J.P.O., and S.K. designed the study. K.W.T., A.K., and S.K. developed the monobodies. K.W.T. and S.T.T. performed biochemical and cell-based characterization. K.W.T., S.T.T., and T.H. determined the crystal structures. K.W.T. and C.F. performed the animal experiments. X.H., C.Y., and Y.Z. performed computational simulations. All authors analyzed data. K.W.T. and S.K. wrote the manuscript from input from all authors.

## Competing interests

K.W.T., A.K., T.H., and S.K. are listed as inventors on a patent application on RAS-targeting monobodies filed by New York University (Application No. 63/121,903). A.K. and S.K. are listed as inventors on issued and pending patents on the monobody technology filed by The University of Chicago (US Patent 9512199 B2 and related pending applications). B.G.N. is a co-founder, chair of the Scientific Advisory Board (SAB), and holds equity in Navire Pharmaceuticals; a co-founder and SAB member and holds equity in and receives consulting fees from Northern Biologics; an SAB member and holds equity in and receives consulting fees from Arvinas; an SAB member and holds equity in Recursion Pharma; a co-founder and holds equity in and receives consulting fees from Jengu Therapeutics; serves as an expert witness for Johnson and Johnson; his spouse owns shares in Amgen. S.K. is an SAB member and holds equity in and receives consulting fees from Black Diamond Therapeutics; receives research funding from Puretech Health and Argenx BVBA. The other authors declare no competing interests.
