## [Peer Review File · Nature Communications]

REVIEWER COMMENTS

Reviewer #1 (Remarks to the Author):

In this manuscript Teng, KW, et al. use phage and yeast display to develop a novel monobody that selectively binds KRAS(G12C) and KRAS(G12V). The monobody, 12VC1, was able to inhibit downstream KRAS signaling both in cells and in vivo. Excitingly, 12VC1 was fused to VHL to induce KRAS mutant degradation in cells and in vivo. This discovery is a major step forward in the development of noncovalent inhibitors/degraders for KRAS mutants and could easily be applied to other “undruggable” proteins.

Teng, KW, et al. used an elegant phage/yeast display system to enrich for monobodies that specifically bound active, GTP-KRAS vs GDP-KRAS. Current covalent inhibitors being investigated clinically bind only inactive, GDP-KRAS. Since most mutants, like G12V, do not rapidly shuttle between GTP and GDP bound these monobodies provide a great tool for studying the inhibition of active KRAS in a noncovalent manner. The authors used BLI to characterize 12VC1 binding to wildtype KRAS, KRAS mutants, and RAS isoforms. This data shows a clear preference for GTP bound, mutant KRAS G12C and G12V over all other forms of the protein.

Next, the authors tested whether 12VC1 could engage KRAS mutants in live cells. Using an EGFP-KRAS overexpression system, they showed clear engagement in colocalization experiments between EGFP-KRAS and mCherry-12VC1. However, in the text these experiments are misleading as it is not clearly stated that an overexpressed, fluorescent KRAS is being used. Next, the authors show clear pulldown of KRAS from cell lysates using a biotinylated 12VC1, with increased pulldown of active KRAS upon EGF stimulation. Mass spectrometry of the elutions from these pulldowns shows very good specificity of 12VC1 for KRAS.

To gain insight into the specificity of 12VC1 binding to KRAS, the authors crystallized 12VC1 in complex with HRASG12C. This revealed key residues on 12VC1 (V33, A48, K50) that recognized the small hydrophobic residues of mutant KRAS G12C and G12V. Mutation of these residues to alanine disrupted binding, supporting the authors model for KRAS recognition. They argue that the shallow pocket between KRAS mutants and 12VC1 would not be filled by the wt KRAS Gly and that other KRAS mutants with larger, charged residues at this position, like G12D, would clash with the key 12VC1 amino acids. Supporting their hypothesis, a non-physiologically relevant G12A KRAS mutant is also bound by 12VC1. Furthermore, a crystal structure of a modified 12VC1 with wt HRAS supports the hypothesis that a shallow pocket recognizes the G12C and G12V mutants.

Next, the authors investigated the in cell activity of 12VC1 in KRAS G12C, G12V, G12D, and wild type cells. As expected, decreases in downstream KRAS signaling were observed in G12C and G12V lines, but not in G12D or wild type lines. This specificity was also seen in proliferation experiments. Excitingly, in vivo efficacy was observed as 12VC1 was able to decrease tumor growth of xenografted PATU9802 cells harboring a G12V mutation. Western blotting did not detect 12VC1 in the remaining PATU9802 tumors suggesting that all cells expressing 12VC1 did not proliferate.

Finally, to add an extra layer of utility to their monobody, the authors conjugated 12VC1 with VHL with hopes of degrading endogenous KRAS G12C and G12V. Some degradation was observed in Rasless MEFs transfected with KRAS G12C. It appears to be on mechanism, however the MLN4924 control for G12C degradation. The authors also tested their monobody conjugate in G12V and G12C cancer cells and observed some degradation, however proteasome inhibitor controls were not conducted. Next, degradation of G12C was compared to inhibition. Interestingly, while pErk signaling recovered after dox removal for the inhibitor, Erk signaling did not recover as significantly after dox removal of the degrader. This suggests that degradation may attenuate signaling to a greater extent, as has been seen previously for other inhibitor/degrader pairs. A similar trend was observed for KRAS G12C levels, which increased overtime for the inhibitor by did not for the degrader. Excitingly, the degrader decreased tumor volume in mice.

Overall, this was a good story of the development and characterization of novel monobodies for KRAS mutants. The selectivity is unprecedented and the crystallographic data will be invaluable to the development of novel KRAS mutant inhibitors. It was exciting to see activity in vivo and, especially with the VHL fusions. However, the data presented here does not clearly show that the in vivo effects of 12VC1.2 were due to degradation and no inhibition. Likewise, some of the degradation data is not as striking as the authors claim in the text. Moreover, many of the findings of the paper are overstated and there is no real mention of the limitations of the technology (i.e. that it has to be genetically encoded). For example, the authors state “This success of a monobody-based degrader not only establishes the feasibility of selectively degrading endogenous RAS mutants,” however a previous publication on KRASG12C degrading PROTACs already established the feasibility of selectively degrading endogenous RAS mutants. Language like this needs to be tempered and the authors need to be transparent about the systems being studied (see major concerns below). Despite these problems, I would recommend this study for publication after the issues outlined below are addressed.

Non-exhaustive list of grammatical and spelling errors:

Line 27 – should say KRAS since only KRAS mutant degradation is shown

Line 31 – Missing the heading of “Introduction”

Line 45-46 – The part of the sentence that reads “which predicts challenges in achieving efficacy by trapping them in the GDP-bound form” sounds awkward and is a run-on sentence. It can be broken into two sentences and reworded.

Line 60 – delete “in” before “engage” and after “RAS”

Line 79 – “warhead” should be pluralized to “warheads”

Line 92 – “mutant” should be pluralized to “mutants”

Line 118 – PDAC stands for pancreatic ductal adenocarcinoma and should be stated in the text

Line 133 – The rhetorical question is not needed and should be deleted

Line 136 – The sentence should say “...because HRAS and KRAS have identical amino acid sequences...”

Line 210 – “tumor” should be pluralized to “tumors”

Line 257 – “suggest” should be pluralized to “suggests”

Line 263 – “mutant” should be pluralized to “mutants”

Line 267 – “study” should be pluralized to “studies”

Line 271 – “play” should be pluralized to “plays”

Line 301 – “potent in” should be changed to “potent at”

Line 317 – “docking” should be changed to “ligase”

Line 328 – “was” should be changed to “were”

Line 333 – “an” should be inserted between “on” and “AKTA”

Line 373 – “cell line” should be pluralized to “cell lines”

Line 419 – “Loading control was” should be pluralized to “Loading controls were”

Line 465 – “identify” and “represent” should be pluralized to “identifies” and “represents”

Line 469 – insert “the” before “pRetro-TetOne”

Line 499 – same as the above comment

Line 520 – “express” should be “expressed”

Line 521 – “cell line” should be pluralized to “cell lines”

Line 523 – “was” should be changed to “were”

Line 541 – insert “a” between “with” and “clear”

The notation of KRAS(G12X) should be standardized. There are a few examples where the parentheses are omitted.

Major issues:

In the introduction there is a major emphasis on engaging and degrading endogenous KRAS mutants. However, EGFP fusion constructs are overexpressed and use for many of the characterization experiments (Figures 1 and 2 in the main text and supplemental figure 2). The use of the EGFP fusion protein should be explicitly stated in the text (as it is in the figures and figure legends) as to not mislead readers.

The MLN4924 control lane in Figure 4a is not convincing. This experiment should be repeated. For a better signal in this experiment a G12C specific antibody could be used. Additionally, MLN4924 and MG132 experiments should be shown for G12V degradation as well.

Only technical replicates for degradation were performed and reported in Figure 4b. It is unclear how many replicates were performed for degradation experiments in the extended data section. These experiments should be repeated so that there is at least a biological replicate of two.

In Figure 4 western blot analysis of KRAS protein levels from tumors treated with 12VC1.2 should be shown. This was shown in Figure 3 for the inhibitor 12VC1, so it is curious as to why the same type of analysis would not be done with the degrader. These data are necessary because they would 1) show in vivo degradation and 2) allow for the authors to really make the claim that 12VC1.2 induced KRAS degradation is leading to decreased tumor size rather than just inhibition of KRAS by 12VC1.2 binding.

The degradation of KRASG12V in Extended Figure 10d is not convincing. Is the quantification of 0.8 from multiple experiments? It would be nice to see more convincing loss of protein along side MLN4924 and MG132 controls. Without this I think the claims that KRASG12V is being degraded should be tempered.

Minor issues:

Line 36 – Should cite “GTP-State-Selective Cyclic Peptide Ligands of K-Ras(G12D) Block Its Interaction with Raf” by Zhang et al when discussing G12D inhibitors. Could also cite “Drugging an undruggable pocket on KRAS” by Kessler et al. that shows a novel, non-covalent RAS inhibitor, albeit it is not selective.

Line 78 – a brief overview of the PROTAC mechanism (binding to POI and E3 ligase to induce ternary complex and ubiquitination) should be given and properly cited for those unfamiliar with the technology

Line 81 – There have been studies with siRNA and microRNA knockdown of specific KRAS mutants (including KRASG12C and KRASG12V that are the focus of this manuscript) that provide insight into the effect that loss of KRAS protein has on cancer cell viability and in turn the therapeutic potential of modulating KRAS protein levels. (e.g. “Knockdown of Oncogenic KRAS in Non-Small Cell Lung Cancers Suppresses Tumor Growth and Sensitizes Tumor Cells to Targeted Therapy” by Sunaga, et al. and “Selective targeting of point-mutated KRAS through artificial microRNAs” by Acunzo, et al.). This should be briefly addressed and these studies cited.

Line 83 – The PROTAC is different from the inhibitor because it is eliminating any potential scaffolding roles for GTP-bound KRAS

Line 255 – The phrase “establishes the feasibility of selectively degrading endogenous RAS mutants” is a bit overstated. KRASG12C targeting PROTACs published earlier this year were the first example showing selective degradation of endogenous RAS mutants and therefore established the feasibility of degrading endogenous RAS proteins. Although covalent and inferior to the parent inhibitors, these PROTACs were also selective for mutant vs wild type KRAS. The language here should be toned down and it should be noted that this work is complementary to the previously published PROTAC report.

Line 281 - “GTP-State-Selective Cyclic Peptide Ligands of K-Ras(G12D) Block Its Interaction with Raf” should be cited here as well

Line 297 – Monobody degraders are not PROTACs. PROTACs by definition are small molecules composed of ligands for an E3 ligase and a POI.

Fig 1c – an insert zooming in on foci showing overlapping fluorescence signal would make it easier to visualize the co-staining

In Figure 1d there should be a panel showing the normalized data to tubulin. The loading, as can be expected with tumor samples, is very inconsistent so it is hard to draw real conclusions from looking at the blot.

Figure 3c – The graph needs a title stating that these data are from PATU8902 xenografts.

Figure 4d – The graph needs a title stating that these data are from H23 xenografts.

Rationale for the 1:1 seeding of 12VC1 and MB(Neg) stable cells in proliferation experiments should be given in the experimental. This should also be present in the body of the manuscript.

What are the prominent proteins in the middle of the gel in Extended Data Figure 3? Is that just keratin or is 12VC1 binding to something other than KRAS in these cells? Was this protein observed in the AP-MS experiments?

In Extended Figure 8c it looks like dox treatment itself reduced tumor size. This should be addressed when this figure is referenced in the body of the main text.

This may have been difficult to do based on the number of mice needed to conduct these studies, but why was only one mutant used in xenograft experiments in figures 3 and figure 4?

Reviewer #2 (Remarks to the Author):

The manuscript entitled 'Selective and non-covalent targeting of RAS mutants for inhibition and degradation' by Teng et al. describes the generation and characterization of a GTP-specific KRAS G12V/C monobody that inhibits RAF-RBD binding and KRAS G12V/C driven signaling. The monobody is also used in conjunction with PROTAC-based technology to promote G12V/C specific RAS degradation. While limitations exist in the expression of/penetration of monobodies for direct therapeutic approaches, the monobodies represent novel tool biologics. Findings herein show novelty in use of 1) monobodies for recognition of mutant-selective KRAS in the active GTP-bound state and 2) PROTAC-based technology for degradation of intracellular RAS. However, some concerns exist with the manuscript in its present form as delineated below.

1. Why is there a significant difference in the K_d (~10-fold) obtained by two different methods, e.g. G12V GTPyS of $K_d = 14 \pm 6.3$ nM or 100 ± 39 nM. Moreover, the K_d error appears high.

2. The immunofluorescence data in Figure 1 requires quantification with regard to RAS mutant status.

3. In Figure 1 and 2, KRAS G12S binds with about 30-fold lower affinity than G12C despite similar side chain size/charge. The crystal structure of HRAS in complex with 12VC1 shows structural evidence for the affinity to KRAS G12C, with the authors stating: "This observation suggested that this pocket 155 directly recognizes small and uncharged side chains at residue 12; by contrast an unfilled 156 pocket, which would occur with wild-type Gly, is energetically unfavorable."

With these observations in mind, are there hypotheses/data as to the discrepancy between KRAS G12C and G12S? Also on Line 152-158: The authors state that Asp12 is bulkier than Cys12 and hence it can't fit in the shallow pocket formed by V33, A48 and K50. However, the molecular volumes of Asp12 (111 Å³) and Cys12 (108.5 Å³) are approximately the same. So, the presence of Asp12 may not significantly destabilize the complex. Moreover, Asp12 has high propensity to form salt-bridge interactions with pocket lining residue Lys50, which may further stabilize the complex.

4. The bonds shown in the Figure 2a (bottom left panel) appear misaligned re: interaction of SWII and G12C. In Figure 2 legend, the structure was solved in GTP-bound form but the nucleotide-bound state is not mentioned.

5. In Figure 3C, it is unclear why tumor growth is higher in 12VC1 -DOX with respect to MB(neg)-Dox. Also, the tumor size after 50 days for 12VC1 -DOX is ~ 450 mm³ while its only ~300 mm³ for MB(neg)-Dox. Some explanation is needed here.

Also, with regard to Figure 3 panels C/D, the authors state: "12VC1 expression significantly reduced tumor growth, whereas expression of MB(Neg) had no impact (Fig. 3c, Extended Data Fig. 8c). We did not detect expression of 12VC1 from these tumors at the end of the xenograft experiment, indicating that the proliferated tumor cells either lost 12VC1 expression due to silencing, or they did not express the monoclonal antibody to begin with, which is a probable scenario, given that a small fraction of cells without monoclonal antibody expression were present in the polyclonal population of the stable cell line (Extended Data Fig. 7c)."

With regard to the authors' hypothesis that the persisting tumor cells in 12VC1-treated mice did not express the monoclonal antibody, it is unclear from the text/methods how long doxycycline was maintained.

Was it a one-time dosage to allow for one instance of monobody expression? Additionally, as differential degradation is a possible explanation for the monobody levels, were experiments comparing the lifetime/expression of the negative monobody control vs. 12VC1 conducted?

6. The authors see that in all tumor samples (regardless of +/- dox, +/- 12VC1) there is a significant decrease in pERK levels as compared to cells at the time of injection. With increases in tumor size/proliferation seen across the board, this trend is perplexing as persistent signaling would be expected. Are trends seen across all tumor lysates, and if so, do the authors have an explanation for this decreased signaling?

7. In Figure 4, it is unclear why RAS ubiquitination was not directly monitored as a direct readout for ubiquitin degradation, as opposed to indirect readouts provided by the investigators. Also, in Figure 4 panel C, a comparison of 12VC1 vs a VHL-fusion of 12VC1 and its effect on H23 signaling is shown. Interestingly, doxycycline removal at 72 hrs allows for rebound in signaling effect/RAS levels for 12VC1 alone, but not the VHL-fusion. Is this due to the sustained expression of VHL-12VC1.2 despite doxycycline removal? Is this observed in other replicates, or do the authors think this persistent expression is due to lifetime of the monobody fusion/biological complexing with KRAS or some other explanation?

8. The data shown in extended Fig. 5a needs further clarification. How did the authors quantify the stability of complex from MD data? What is the rationale or metric for complex stability? It would be helpful to quantify the interaction energy between RAS mutants and monobody using MMPBSA or MMGBSA methods.

9. In extended data Fig. 6, the RMSD of cluster-3 with regard to both WT and G12C starting structures is approximately 3.2 Å. However, RMSD profiles show the structural deviation of WT and G12C below 2Å.

10. In Extended Data Figure 10 Panel C, more clarification/additional labeling is required. Are the first grouping of 3 timepoints for KRAS WT in RASless MEFs or KRAS G12C? With the current labeling here (and in general for Extended Figure 10/11), it is unclear how some of these experiments differ in design from those in the main text and why some trends (RAS levels, pERK/ERK) are different with some of the supplemental data.

Reviewer #3 (Remarks to the Author):

This work builds on previous efforts by the authors to discover monobodies that can interact with Ras. The work is sufficiently novel to warrant publication in Nature Communications. Specifically, the discovery biochemical, structural and functional characterization of small proteins that can discriminate between WT and mutant Ras to block signaling and the complementary idea of coupling these selective agents to E3 ligases (VHL in this case) to trigger mutant-specific degradation are interesting and timely contributions to the field. In addition, the manuscript is well-written and the experiments appear carefully done. I have a few minor queries or suggestions that the authors might want to consider more as food for thought rather than as required changes prior to publication.

1) In Figure 1d (and, in general many of the western blots), it is confusing as to which tag was used for the IP versus the western blot. It would be easier for the reader if this information was added to the figure.

2) Please add the resolution of the structures either to the main methods or to the figure legend.

3) Line 106: Consider explaining that monobodies are FnIII-derived scaffolds for readers that are less familiar with the authors' previous work.

4) Line 92: Consider describing this work as the “discovery” of a noncovalent inhibitor of Ras rather than “development” which, at least for some scientists, implies clinical evaluation of a potential therapeutic.

5) Line 152 Consider explaining what “computational structural analysis” was used to reveal the shallow pocket (visual inspection in PyMol/Coot? Or a specific program?) More generally, the authors rule out a change in backbone conformation being the cause of 12VC1 mutant selectivity. How much backbone change is there between the wild type and mutant forms of Ras in the absence of monobodies?

6) Figure 4b: Quantification of how much mutant Ras (rather than total Ras) is degraded by the monobody-VHL fusion would strengthen this section. More generally, while the monobody-VHL fusion is very clever and, as the authors point out, is potentially a great biological tool to more generally evaluate the potential of “degraders” against a given target without having to invest in the discovery of small molecule ligands, it isn't clear from comparison of the tumor efficacy data in

figures 3 and figure 4 whether the monobody-VHL fusion has any greater effect than the inhibitory monobody itself in this case.

7) Extended data figure 10: In Figure 10c, Ras degradation seems less profound than in Figure 4b. What is the difference between these experiments? Presumably in the RasLess MEFs, there is only mutant Ras? Did the authors measure the on and off rates of 12VC1.1 and 12VC1.2? Could this possibly explain the differences in efficacy?

8) Figure 3d – why is pERK down so much in all tumor samples, even when most of them don't show monobody expression (at least in the surviving tumor cells at the end of the experiment)? pERK levels in this blot aren't addressed in the text.

Response to the Reviewers' Comments

Title: Selective and Noncovalent Targeting of RAS Mutants for Inhibition and Degradation

Authors' responses are in blue. The Extended Data section referred to by the reviewers has been re-named as Supporting Information per formatting requirements for Nature Communications.

REVIEWER COMMENTS

Reviewer #1 (Remarks to the Author):

In this manuscript Teng, KW, et al. use phage and yeast display to develop a novel monobody that selectively binds KRAS(G12C) and KRAS(G12V). The monobody, 12VC1, was able to inhibit downstream KRAS signaling both in cells and in vivo. Excitingly, 12VC1 was fused to VHL to induce KRAS mutant degradation in cells and in vivo. This discovery is a major step forward in the development of noncovalent inhibitors/degraders for KRAS mutants and could easily be applied to other "undruggable" proteins.

Teng, KW, et al. used an elegant phage/yeast display system to enrich for monobodies that specifically bound active, GTP-KRAS vs GDP-KRAS. Current covalent inhibitors being investigated clinically bind only inactive, GDP-KRAS. Since most mutants, like G12V, do not rapidly shuttle between GTP and GDP bound these monobodies provide a great tool for studying the inhibition of active KRAS in a noncovalent manner. The authors used BLI to characterize 12VC1 binding to wildtype KRAS, KRAS mutants, and RAS isoforms. This data shows a clear preference for GTP bound, mutant KRAS G12C and G12V over all other forms of the protein.

Next, the authors tested whether 12VC1 could engage KRAS mutants in live cells. Using an EGFP-KRAS overexpression system, they showed clear engagement in colocalization experiments between EGFP-KRAS and mCherry-12VC1. However, in the text these experiments are misleading as it is not clearly stated that an overexpressed, fluorescent KRAS is being used. Next, the authors show clear pulldown of KRAS from cell lysates using a biotinylated 12VC1, with increased pulldown of active KRAS upon EGF stimulation. Mass spectrometry of the elutions from these pulldowns shows very good specificity of 12VC1 for KRAS.

To gain insight into the specificity of 12VC1 binding to KRAS, the authors crystallized 12VC1 in complex with HRASG12C. This revealed key residues on 12VC1 (V33, A48, K50) that recognized the small hydrophobic residues of mutant KRAS G12C and G12V. Mutation of these residues to alanine disrupted binding, supporting the authors model for KRAS recognition. They argue that the shallow pocket between KRAS mutants and 12VC1 would not be filled by the wt KRAS Gly and that other KRAS mutants with larger, charged residues at this position, like G12D, would clash with the key 12VC1 amino acids. Supporting their hypothesis, a non-physiologically relevant G12A KRAS mutant is also bound by 12VC1. Furthermore, a crystal structure of a modified 12VC1 with wt HRAS supports the hypothesis that a shallow pocket recognizes the G12C and G12V mutants.

Next, the authors investigated the in cell activity of 12VC1 in KRAS G12C, G12V, G12D, and wild type cells. As expected, decreases in downstream KRAS signaling were observed in G12C and G12V lines, but not in G12D or wild type lines. This specificity was also seen in proliferation experiments. Excitingly, in vivo efficacy was observed as 12VC1 was able to decrease tumor growth of xenografted PATU9802 cells harboring a G12V mutation. Western blotting did not detect 12VC1 in the remaining PATU9802 tumors suggesting that all cells expressing 12VC1 did not proliferate.

Finally, to add an extra layer of utility to their monobody, the authors conjugated 12VC1 with VHL with hopes of degrading endogenous KRAS G12C and G12V. Some degradation was observed in Rasless MEFs transfected with KRAS G12C. It appears to be on mechanism, however the MLN4924 control for G12C degradation. The authors also tested their monobody conjugate in G12V and G12C cancer cells and observed some degradation, however proteasome inhibitor controls were not conducted. Next, degradation of G12C was compared to inhibition. Interestingly, while pErk signaling recovered after dox removal for the inhibitor, Erk signaling did not recover as significantly after dox removal of the degrader. This suggests that degradation may attenuate signaling to a greater extent, as has been seen previously for other inhibitor/degrader pairs. A similar trend was observed for KRAS G12C levels, which increased overtime for the inhibitor but did not for the degrader. Excitingly, the degrader decreased tumor volume in mice.

Overall, this was a good story of the development and characterization of novel monobodies for KRAS mutants. The selectivity is unprecedented and the crystallographic data will be invaluable to the development of novel KRAS mutant inhibitors. It was exciting to see activity in vivo and, especially with the VHL fusions. However, the data presented here does not clearly show that the in vivo effects of 12VC1.2 were due to degradation and not inhibition. Likewise, some of the degradation data is not as striking as the authors claim in the text. Moreover, many of the findings of the paper are overstated and there is no real mention of the limitations of the technology (i.e. that it has to be genetically encoded). For example, the authors state "This success of a monobody-based degrader not only establishes the feasibility of selectively degrading endogenous RAS mutants," however a previous publication on KRASG12C degrading PROTACs already established the feasibility of selectively degrading endogenous RAS mutants. Language like this needs to be tempered and the authors need to be transparent about the systems being studied (see major concerns below). Despite these problems, I would recommend this study for publication after the issues outlined below are addressed.

We appreciate the reviewer's careful examination of our manuscript, enthusiasm for our work, and recognition of the unprecedented selectivity of our newly developed monobody 12VC1. We agree with the reviewer that monobody 12VC1 could enable a major step towards developing noncovalent inhibitors against RAS mutants. We share the reviewer's excitement over the aspect of degrading RAS mutant selectively using the monobody technology. We also agree with the reviewer that, despite the overwhelming positivity, we should highlight some of the limitations of the monobody technology in cell-based assays, as well as temper down some of the statements, given that the druggability and selectivity towards KRAS mutant have rapidly evolved within the last few years. Descriptions of the limitations of our work have been included in the results and discussion sections (L.235-240 and L.350-355).

Non-exhaustive list of grammatical and spelling errors:

Line 27 – should say KRAS since only KRAS mutant degradation is shown

Line 31 – Missing the heading of “Introduction”

Line 45-46 – The part of the sentence that reads “which predicts challenges in achieving efficacy by trapping them in the GDP-bound form” sounds awkward and is a run-on sentence. It can be broken into two sentences and reworded.

Line 60 – delete “in” before “engage” and after “RAS”

Line 79 – “warhead” should be pluralized to “warheads”

Line 92 – “mutant” should be pluralized to “mutants”

Line 118 – PDAC stands for pancreatic ductal adenocarcinoma and should be stated in the text

Line 133 – The rhetorical question is not needed and should be deleted

Line 136 – The sentence should say “...because HRAS and KRAS have identical amino acid sequences...”

Line 210 – “tumor” should be pluralized to “tumors”

Line 257 – “suggest” should be pluralized to “suggests”

Line 263 – “mutant” should be pluralized to “mutants”

Line 267 – “study” should be pluralized to “studies”

Line 271 – “play” should be pluralized to “plays”

Line 301 – “potent in” should be changed to “potent at”

Line 317 – “docking” should be changed to “ligase”

Line 328 – “was” should be changed to “were”

Line 333 – “an” should be inserted between “on” and “AKTA”

Line 373 – “cell line” should be pluralized to “cell lines”

Line 419 – “Loading control was” should be pluralized to “Loading controls were”

Line 465 – “identify” and “represent” should be pluralized to “identifies” and “represents”

Line 469 – insert “the” before “pRetro-TetOne”

Line 499 – same as the above comment

Line 520 – “express” should be “expressed”

Line 521 – “cell line” should be pluralized to “cell lines”

Line 523 – “was” should be changed to “were”

Line 541 – insert “a” between “with” and “clear”

The notation of KRAS(G12X) should be standardized. There are a few examples where the parentheses are omitted.

We really appreciate that the reviewer had carefully read the manuscript and pointed out these mistakes and suggestions. We have fixed them and proofread our revised manuscript.

We have numbered the reviewer’s comments below for ease of referencing.

Major issues:

1. In the introduction there is a major emphasis on engaging and degrading endogenous KRAS mutants. However, EGFP fusion constructs are overexpressed and used for many of the characterization experiments (Figures 1 and 2 in the main text and supplemental figure 2). The use of the EGFP fusion protein should be explicitly stated in the text (as it is in the figures and figure legends) as to not mislead readers.

We agree with the reviewer that we should clearly distinguish data sets acquired with overexpressed and those with endogenous KRAS. The EGFP fusion constructs were initially used for facilitating validation of the monobody inhibitors. We have, however, followed up these initial experiments with experiments using endogenously expressed, untagged RAS molecules. We have explicitly stated the use of EGFP fusion protein in the main text and figure legends.

2. The MLN4924 control lane in Figure 4a is not convincing. This experiment should be repeated. For a better signal in this experiment a G12C specific antibody could be used. Additionally, MLN4924 and MG132 experiments should be shown for G12V degradation as well.

We realized that our description of this experiment was incomplete and confusing. In this experiment, we first expressed the monobody degrader for 24 hours in order to accumulate it at a sufficient level and then treated the cells with the proteasome inhibitors for an additional 24 hours. The data showed no further degradation of RAS upon proteasome inhibitor treatment, hence the expression level of RAS at 48 hours (i.e. after 24 hours of proteasome inhibitor treatment) matched with that observed after 24 hours of MB expression (i.e. at the start of proteasome inhibitor treatment). As one can see from the pan-RAS blotting panels (Fig. 4a), it takes the VHL-monobody (MB) fusion 48 hours to significantly deplete RAS mutants in cells. However, we needed to limit the length of the inhibitor treatment to 24 hours, because these inhibitors are themselves toxic. We devised this experimental scheme to accommodate these conflicting requirements.

We have also repeated this experiment as the reviewer suggested, but this time we added the inhibitor earlier, after just 8 hours of VHL-MB expression for a total expression time of 24 hours. The degradation of RAS was rescued completely with the inhibitor treatment. The expression levels of RAS with either of the inhibitors were significantly higher than without the inhibitors at 24 hours (see Supplementary Fig. 10e, pasted below).

We recognized that this experimental scheme was not properly conveyed in the original figure legend. We have improved the description and added schemes to show the experimental designs (Fig. 4 and Supplementary Fig. 10e).

For G12V, we have now included the inhibitor treatments for PATU8902 cells expressing the VHL-MB fusions (Fig. 4b), which also show inhibition of degradation.

Fig. 4a

Supplementary Fig. 10e

Fig. 4b

3. Only technical replicates for degradation were performed and reported in Figure 4b. It is unclear how many replicates were performed for degradation experiments in the extended data section. These experiments should be repeated so that there is at least a biological replicate of two.

We thank the reviewer for pointing this out. We have repeated two more biological replicates of the experiment stated in 4b (see data from our response to point #2 above). The experiments in the Supplementary section has two or more biological replicates. We have ensured that we state the number of biological replicates for the experiments in the Supplementary section.

4. In Figure 4 western blot analysis of KRAS protein levels from tumors treated with 12VC1.2 should be shown. This was shown in Figure 3 for the inhibitor 12VC1, so it is curious as to why the same type of analysis would not be done with the degrader. These data are necessary because they would 1) show in vivo degradation and 2) allow for the authors to really make the claim that 12VC1.2 induced KRAS degradation is leading to decreased tumor size rather than just inhibition of KRAS by 12VC1.2 binding.

We have included the analysis of KRAS protein levels from these tumor lysates. These data show that the VHL-12VC1.2 fusion decreased the KRAS levels in the +dox samples, whereas the VHL-MB(Neg) fusion did not. Please note that the extra bands at 25 kDa and above in the KRAS blotting and those at 50 kDa in the HA blotting are artifacts of using an anti-mouse secondary antibody on samples of mouse origin (likely the immunoglobulin light and heavy chains). The use of an anti-mouse secondary antibody in these blots was necessary, because the only KRAS-specific antibody that has been validated in the literature for Western blotting was a mouse antibody. We have also included analysis of the total RAS levels using a rabbit antibody, which showed a much weaker band at 25 kDa, confirming that the band is an artifact rather than RAS.

Supplementary Fig. 11b

We would also like to emphasize that these xenograft experiments were performed using a polyclonal population of transduced cells that showed diverse expression levels of the VHL-MB fusion including a low percentage of non-expressers. Over the duration of dox treatment, non-expressers and low-expressers within this polyclonal population can continue to proliferate. Hence, the pERK level and RAS levels at the end of the xenograft experiment, which integrate the levels over these heterogeneous cells, may underestimate the impact of monobody expression on RAS signaling and degradation. We have added a description of this point (L. 350-355).

5. The degradation of KRASG12V in Extended Figure 10d is not convincing. Is the quantification of 0.8 from multiple experiments? It would be nice to see more convincing loss of protein along side MLN4924 and MG132 controls. Without this I think the claims that KRASG12V is being degraded should be tempered.

As described in our response to comment 2 above, we have repeated (n=2 biological replicate) the experiment of PATU8902 with the KRAS(G12V) mutant status, in addition to the experimental data included in the original manuscript (Fig. 4b and Supplementary Fig. 10d). Results from these experiments show that 12VC1.1 is superior in degrading RAS to 12VC1.2. The new data include KRAS levels probed using a KRAS-selective antibody, which consistently show a greater reduction of the KRAS level with 12VC1.1 than with 12VC1.2 (see the pasted figures below). Please note that PATU8902 has a heterozygous KRAS mutation status and still contains wildtype KRAS. Thus, the complete depletion of KRAS is not expected even when the VHL-MB fusion completely degraded KRAS(G12V). We have also included MLN4924 and MG132 inhibitor controls as suggested by the reviewer, which show inhibition of degradation.

Minor issues:

6. Line 36 – Should cite “GTP-State-Selective Cyclic Peptide Ligands of K-Ras(G12D) Block Its Interaction with Raf” by Zhang et al when discussing G12D inhibitors. Could also cite “Drugging an undruggable pocket on KRAS” by Kessler et al. that shows a novel, non-covalent RAS inhibitor, albeit it is not selective.

We thank the reviewer for bringing these fantastic studies to our attention. We have cited them in our revised manuscript (Main Text, Ref #16).

7. Line 78 – a brief overview of the PROTAC mechanism (binding to POI and E3 ligase to induce ternary complex and ubiquitination) should be given and properly cited for those unfamiliar with the technology

We agree with the reviewer. We have included a brief overview of the PROTAC mechanism and more references (Main Text, Ref#21, 22).

8. Line 81 – There have been studies with siRNA and microRNA knockdown of specific KRAS mutants (including KRASG12C and KRASG12V that are the focus of this manuscript) that provide insight into the effect that loss of KRAS protein has on cancer cell viability and in turn the therapeutic potential of modulating KRAS protein levels.

(e.g. “Knockdown of Oncogenic KRAS in Non-Small Cell Lung Cancers Suppresses Tumor Growth and Sensitizes Tumor Cells to Targeted Therapy” by Sunaga, et al. and “Selective targeting of point-mutated KRAS through artificial microRNAs” by Acunzo, et al.). This should be briefly addressed and these studies cited.

We appreciate the reviewer for pointing out these references, we have included them in our revised manuscript (Main Text, Ref 3, 4).

9. Line 83 – The PROTAC is different from the inhibitor because it is eliminating any potential scaffolding roles for GTP-bound KRAS

Thank you for raising this excellent point. We have stated it in the revised manuscript (L. 91-94).

10. Line 255 – The phrase “establishes the feasibility of selectively degrading endogenous RAS mutants” is a bit overstated. KRASG12C targeting PROTACs published earlier this year were the first example showing selective degradation of endogenous RAS mutants and therefore established the feasibility of degrading endogenous RAS proteins.

Although covalent and inferior to the parent inhibitors, these PROTACs were also selective for mutant vs wild type KRAS. The language here should be toned down and it should be noted that this work is complementary to the previously published PROTAC report.

We agree with the reviewer that our work complements other studies in the field of RAS drug discovery, and we have tempered down our language in the revised manuscript. We recognize the impact of PROTACs based on a covalent inhibitor, as a proof of concept demonstrating that a RAS mutant can be selectively degraded. Admittedly the publication of the mentioned work while this manuscript was under review for different journals has taken some novelty away from our study. However, we are still confident that mutant-selective, noncovalent degradation as demonstrated in our work, is significantly novel in terms of addressing the RAS feedback mechanism and the ability to target KRAS G12V and other RAS mutants.

11. Line 281 - “GTP-State-Selective Cyclic Peptide Ligands of K-Ras(G12D) Block Its Interaction with Raf” should be cited here as well

We thank the review for pointing this out. We have cited this paper here as well.

12. Line 297 – Monobody degraders are not PROTACs. PROTACs by definition are small molecules composed of ligands for an E3 ligase and a POI.

Our interpretation of the definition of PROTAC is from the 2001 PNAS article by Sakamoto et al., “Protacs: chimeric molecules that target proteins to the Skp1-Cullin-F box complex for ubiquitination and degradation” where PROTAC was first mentioned. PROTAC refers generally to any bi-functional molecule that directs a target protein to an E3 ligase for ubiquitylation and degradation. The first PROTAC did indeed utilized small molecule, but the description of what a PROTAC is was more generalized. We acknowledge that the reviewer suggests that we differentiate protein based PROTAC with small molecule based PROTAC to highlight the difference in utility, e.g. one is cell permeable and drug-like. We believe that an essential point here is to clearly distinguish protein-based and small molecule-based PROTACs and state their pros and cons. We have expanded these descriptions by referring to protein based PROTAC as PROTAC-like degraders.

13. Fig 1c – an insert zooming in on foci showing overlapping fluorescence signal would make it easier to visualize the co-staining

We thank the reviewer for making this suggestion, we have included the fluorescence intensity profiles of the regions across the cells from GFP and mCherry fluorescence channels (Fig. 1c, see below) to help the reader visualize co-localization of the fluorescence signals. These profiles clearly show colocalization for KRAS(G12C) and KRAS(G12V).

**Fig. 1c**

14. In Figure 1d there should be a panel showing the normalized data to tubulin. The loading, as can be expected with tumor samples, is very inconsistent so it is hard to draw real conclusions from looking at the blot.

We thank the reviewer for pointing this out. We have normalized the pull-down band intensities to the total RAS levels of the input lysates, and provided quantification below the captured RAS bands.

**Fig. 1d**

15. Figure 3c – The graph needs a title stating that these data are from PATU8902 xenografts.

We have added a title in the revised manuscript.

16. Figure 4d – The graph needs a title stating that these data are from H23 xenografts.

We have added a title.

17. Rationale for the 1:1 seeding of 12VC1 and MB(Neg) stable cells in proliferation experiments should be given in the experimental. This should also be present in the body of the manuscript.

We assume that this comment refers to the proliferation experiment in Figure 3b. We have stated the rationale of why we start the seeding of 12VC1 expressing cells with wild-type untransduced cells in the main text (L. 207-210). We believe that this design produces lower experimental variations than monitoring cell growth in separate cultures.

18. What are the prominent proteins in the middle of the gel in Extended Data Figure 3? Is that just keratin or is 12VC1 binding to something other than KRAS in these cells? Was this protein observed in the AP-MS experiments?

We appreciate the reviewer for careful examination of our data. It is most likely to be bovine serum albumin (BSA) included in the wash buffers. BSA and other contaminants in this molecular weight range (e.g. keratin type I and II) were detected in the AP-MS experiments.

19. In Extended Figure 8c it looks like dox treatment itself reduced tumor size. This should be addressed when this figure is referenced in the body of the main text.

The tumor weights were quantified (Fig. 3e), which did not show a significant difference between +/- dox treatment. The sizes of tumors generated from MB(Neg) varied greatly, and we agree with the reviewer that one may have to include more mice to determine whether dox treatment itself had a significant effect. Our cell-based data show (Supplementary Fig. 8b) that the expression of MB(Neg) in PATU8902 cells does have a small effect after a long period of time. While the influence of dox can be difficult to see between the MB(Neg) tumors, the 12VC1 tumors with dox treatments were all much smaller across the board compared to 12VC1 tumors without dox treatments. We now discuss this point in L. 235-245.

Supplementary Fig. 8b

20. This may have been difficult to do based on the number of mice needed to conduct these studies, but why was only one mutant used in xenograft experiments in figures 3 and figure 4? We used only one mutant per xenograft, because, as the reviewer correctly imagined, we tried to balance between minimizing the number of mice we needed to use in the study and demonstrating efficacy and novelty of our approaches. We showed that 12VC1 as an inhibitor worked potently in cell-based assay against cancer cell lines harboring KRAS(G12C) mutation, and we subsequently showed that it achieved similar potency in signaling assay and cell viability assay compared with the covalent inhibitor (Supplementary Fig. 9). Because of extensive studies documenting successes of inhibitors against G12C in the literature, we designed the xenograft experiment using 12VC1 as the inhibitor to demonstrate its efficacy against a tumor harboring G12V mutation, an unmet challenge in RAS drug discovery.

We designed the xenograft experiment with the VHL-MB fusion to demonstrate its efficacy against cancer cells, H23, that are not very responsive to the covalent inhibitors and also can be transduced at a high level. As the reviewer correctly pointed out, these reagents must be introduced in cells via viral transduction, which always leaves a substantial fraction of population of transduced cells expresses low to no levels of a monoclonal antibody inhibitor or degraders that compromises the assessment of the efficacy of an inhibitor or degrader (see our response to point #2 above). We have included descriptions of our rationales for the design of these experiments (L. 226-229 and L. 309-313).

Reviewer #2 (Remarks to the Author):

The manuscript entitled 'Selective and non-covalent targeting of RAS mutants for inhibition and degradation' by Teng et al. describes the generation and characterization of a GTP-specific KRAS G12V/C monoclonal antibody that inhibits RAF-RBD binding and KRAS G12V/C driven signaling. The monoclonal antibody is also used in conjunction with PROTAC-based technology to promote G12V/C specific RAS degradation. While limitations exist in the expression of/penetration of monoclonal antibodies for direct therapeutic approaches, the monoclonal antibodies represent novel tool biologics. Findings herein show novelty in use of 1) monoclonal antibodies for recognition of mutant-selective KRAS in the active GTP-bound state and 2) PROTAC-based technology for degradation of intracellular RAS. However, some concerns exist with the manuscript in its present form as delineated below.

We thank the reviewer for recognizing the novelty of 12VC1 as an important tool biologic for selective targeting of RAS mutants. We thank the reviewer for the constructive criticism and helpful suggestions.

1. Why is there a significant difference in the K_d (~10-fold) obtained by two different methods, e.g. G12V GTPyS of $K_d = 14 \pm 6.3$ nM or 100 ± 39 nM. Moreover, the K_d error appears high.

We thank the reviewer for raising the issue of the difference in affinity between yeast and BLI measurements. The yeast display binding measurement is convenient to perform, because it is a part of our workflow of developing monoclonal antibodies. However, this assay is not a true equilibrium binding experiment and consequently one can determine only apparent K_D values. In addition,

the exact nature of the monobody presented on the yeast surface is unknown. In contrast, biolayer interferometry (BLI) uses purified monobody and RAS samples in a well-defined experimental setting, and it can give the true K_D values. We have measured K_D values of many binding proteins using both methods, and although we see differences in the absolute values in the (apparent) K_D value between the two methods, the rank order among clones is usually conserved. As such yeast surface display is a useful technique. It was indeed confusing to display titration data with two different methods in the same figure. Therefore, we now only show a single-point binding data using yeast display in Figure 1 and moved yeast display titration data to Supplementary Fig. 1. To avoid confusion and to differentiate the two measurements, we have termed K_D measured by yeast display apparent K_D ($K_{D\text{ app}}$).

The errors in the K_D values are the s.d. of independent technical replicates as we state, rather than fitting errors from a single titration that many studies report. We believe that our reported errors are reasonable and represent a realistic level of precision of these methods.

2. The immunofluorescence data in Figure 1 requires quantification with regard to RAS mutant status.

We infer from the reviewer that he/she is referring to the confocal fluorescence images of the cells, rather than immunofluorescence. We have quantified the fluorescence intensity level of the mCherry and EGFP over the surface of the cell, making colocalization easier to observe in our revised manuscript (Fig. 1c). Please also see our response to Reviewer 1, comment 13 above.

3. In Figure 1 and 2, KRAS G12S binds with about 30-fold lower affinity than G12C despite similar side chain size/charge. The crystal structure of HRAS in complex with 12VC1 shows structural evidence for the affinity to KRAS G12C, with the authors stating: "This observation suggested that this pocket 155 directly recognizes small and uncharged side chains at residue 12; by contrast an unfilled 156 pocket, which would occur with wild-type Gly, is energetically unfavorable."

With these observations in mind, are there hypotheses/data as to the discrepancy between KRAS G12C and G12S? Also on Line 152-158: The authors state that Asp12 is bulkier than Cys12 and hence it can't fit in the shallow pocket formed by V33, A48 and K50. However, the molecular volumes of Asp12 (111 Å³) and Cys12 (108.5 Å³) are approximately the same. So, the presence of Asp12 may not significantly destabilize the complex. Moreover, Asp12 has high propensity to form salt-bridge interactions with pocket lining residue Lys50, which may further stabilize the complex.

We thank the reviewer for pointing out our too simplistic statement. We did not imply that the sizes of the side chain and affinity should be linearly correlated. Small differences in the side chain size and/or chemistry may substantially affect the interaction. In addition to being significantly larger than the Ser side chain (~30 versus ~50 x 10⁻²⁴ cm³ according to ref 29), the Cys side chain is more hydrophobic, making it more energetically favorable to be buried in the pocket. Regarding the difference between Cys vs Asp, Asp is much more electronegative

than G12C and incurs greater desolvation penalty upon complex formation. We have included these more nuanced descriptions on page 8 (L. 170-179).

29. Hackel, M., Hinz, H. J. & Hedwig, G. R. Partial molar volumes of proteins: amino acid side chain contributions derived from the partial molar volumes of some tripeptides over the temperature range 10-90 degrees C. *Biophys Chem* **82**, 35-50, doi:10.1016/s0301-4622(99)00104-0 (1999).

4. The bonds shown in the Figure 2a (bottom left panel) appear misaligned re: interaction of SWII and G12C. In Figure 2 legend, the structure was solved in GTP-bound form but the nucleotide-bound state is not mentioned.

We thank the reviewer for noticing that the bond appeared misaligned. The problem was that only one end of the bonds was drawn as stick models, whereas the other end is the backbone of the monobody, which was drawn as a cartoon model. We have revised the graphics so that the interaction is properly displayed (see below).

Fig. 2a

5. In Figure 3C, it is unclear why tumor growth is higher in 12VC1 -DOX with respect to MB(neg)-dox. Also, the tumor size after 50 days for 12VC1 -DOX is ~ 450 mm³ while its only ~300 mm³ for MB(neg)-dox. Some explanation is needed here.

We thank the reviewer for noticing this difference and agree with the reviewer that some clarification is needed to explain this difference. We think that there are two possible explanations. First, although the two stable cell lines were generated from the same cell line, the integration of the retroviral vector can occur at different locations, which somehow made MB(Neg)-expressing cells grow slower. Second, technical variability in tumor injection, including cell harvesting, starting cell number, matrigel mixing, etc. caused different growth efficiency of the tumors between the two groups. We did not observe a difference in growth speed between uninduced MB(Neg) and 12VC1 tumors in our second xenograft experiment conducted using the VHL MB constructs, which utilized the same retroviral vector system. Therefore, it is more likely that technical variability contributed to the growth of MB(Neg) tumors to lag behind 12VC1 tumors. Nonetheless, we are confident that the results support the conclusion that the expression of MB(Neg) did not impact the tumor growth, because the tumor size did not significantly vary with and without dox and also because we observed expression of MB(Neg) in the dox-fed mice. We have noted this point in page 10-11 (L. 235-245).

Also, with regard to Figure 3 panels C/D, the authors state: “12VC1 expression significantly reduced tumor growth, whereas expression of MB(Neg) had no impact (Fig. 3c, Extended Data Fig. 8c). We did not detect expression of 12VC1 from these tumors at the end of the xenograft experiment, indicating that the proliferated tumor cells either lost 12VC1 expression due to silencing, or they did not express the monobody to begin with, which is a probable scenario, given that a small fraction of cells without monobody expression were present in the polyclonal population of the stable cell line (Extended Data Fig. 7c).” With regard to the authors’ hypothesis that the persisting tumor cells in 12VC1-treated mice did not express the monobody, it is unclear from the text/methods how long doxycycline was maintained. Was it a one-time dosage to allow for one instance of monobody expression? Additionally, as differential degradation is a possible explanation for the monobody levels, were experiments comparing the lifetime/expression of the negative monobody control vs. 12VC1 conducted?

We have clarified the dosing method of dox. dox was maintained as feeds from the indicated start date through the end of the experiment. The dox feeds were refreshed weekly. Under these conditions, the expression of monobodies should be fairly constant throughout the experiment if the growth of the cells were not impacted by the monobody expression. We have shown that the percentages of cells expressing fluorescent protein fused monobodies, whether it is MB(Neg) or 12VC1, are fairly constant in cells whose growth is not dependent on KRAS(G12C) or KRAS(G12V) and hence whose growth is not impacted by the monobodies, HEK293T and A375, respectively, over a period of 8-9 days after induction (see below, and Supplementary Fig. 8b). The expression levels of both monobody constructs in these cell lines remained constant over time, as measured by mean fluorescence intensity (MFI) levels. Therefore, these data support the view that selective pressure against cells containing KRAS(G12C) or (G12V) mutants, rather than rapid decay in monobody constructs in cells, is the more likely cause for diminishing monobody expression in tumors. We have described this point in the main text (L. 246-256).

6. The authors see that in all tumor samples (regardless of +/- dox, +/- 12VC1) there is a significant decrease in pERK levels as compared to cells at the time of injection. With increases in tumor size/proliferation seen across the board, this trend is perplexing as persistent signaling would be expected. Are trends seen across all tumor lysates, and if so, do the authors have an explanation for this decreased signaling?

We thank the reviewer for noticing this tendency. The absolute pERK level decreased across the board in this figure. However, the total ERK level as well as the tubulin level was also

reduced. Although we tried to ensure loading equal amount of protein in each lane, difficulties in protein quantification from mouse lysate and cell lysate could have attributed to this uneven loading. We have now normalized the pERK levels to the total ERK levels (Figure 3d). We excluded from quantification those lanes that show tubulin levels that are too low to be quantified. The normalized pERK levels are not dramatically decreased as the absolute band intensities might suggest.

Fig. 3d

7. In Figure 4, it is unclear why RAS ubiquitination was not directly monitored as a direct readout for ubiquitin degradation, as opposed to indirect readouts provided by the investigators. Also, in Figure 4 panel C, a comparison of 12VC1 vs a VHL-fusion of 12VC1 and its effect on H23 signaling is shown. Interestingly, doxycycline removal at 72 hrs allows for rebound in signaling effect/RAS levels for 12VC1 alone, but not the VHL-fusion. Is this due to the sustained expression of VHL-12VC1.2 despite doxycycline removal? Is this observed in other replicates, or do the authors think this persistent expression is due to lifetime of the monobody fusion/biological complexing with KRAS or some other explanation?

The reviewer brought up a good point. Ubiquitination of endogenous RAS was not directly monitored in this work. We have attempted to detect ubiquitinated endogenous RAS through western blot, but it was unsuccessful. We think that ubiquitinated endogenous RAS does not accumulate in the cell abundantly enough to be observed on immunoblots. Consistent with this view, none of the recent papers that investigated RAS degradation by hijacking E3 ligase (Röth et al. Cell Chem. Biol. 2020, Bery et al. Nat. Comm. 2020, Bond et al. ACS Central Sci. 2020) show direct detection of ubiquitinated endogenous RAS. In another recent work, it was also reported that direct observation of ubiquitinated RIT1, a small GTPase closely related to RAS, was not detectable by conventional means, despite RIT1 being degraded via LTZR1-mediated ubiquitylation (Castel et al. Science 2020). Therefore, we decided to include controls with neddylation and proteasome inhibitors instead of pursuing direct detection of poly-ubiquitinated RAS.

We consistently observed that rebounding of pERK was less for cells expressing VHL-MB than for cells expressing inhibitors. The data shown here were representative results from biological replicate of three, as stated in the text.

As the reviewer noted, VHL-12VC1.2 (degrader) was sustainably expressed after dox removal whereas 12VC1 (inhibitor) level declined, although the inhibitor was present at a greater concentration. Perhaps more important, the increases of KRAS and pan-RAS levels with the inhibitor are detectable at the 48 hr point, even before dox removal, when the inhibitor concentration was very high. In contrast, the degrader kept the KRAS level low. The difference in pERK rebounding is undoubtedly caused by a combination of these multiple factors, which together suggest potential advantages of degradation over inhibition. We have described these points in page 13 (L. 300-306).

8. The data shown in extended Fig. 5a needs further clarification. How did the authors quantify the stability of complex from MD data? What is the rationale or metric for complex stability? It would be helpful to quantify the interaction energy between RAS mutants and monobody using MMPBSA or MMGBSA methods.

To further clarify, we measured the stability of complex using following metrics: (1) Average distance between #12 residue in RAS and V33/A48/K50 in monobody; (2) Average RMSD of monobody during MD simulation, with complex aligned based on RAS protein; (3) Average RMSD of loop(G44/A45/F46) of monobody during MD simulation, with complex aligned based on RAS protein; (4) Average distance between K117 in RAS and G44 in monobody. These metrics present the key interactions between RAS and monobody. We have described these points in the Methods section. At present, accurate quantitative calculations of the binding energies for protein-protein complexes are still very challenging for MMPBSA or MMGBSA methods (Chem. Rev. 2019, 119, 16, 9478–9508) and thus we think that quantification using these methods would not be helpful.

9. In extended data Fig. 6, the RMSD of cluster-3 with regard to both WT and G12C starting structures is approximately 3.2 Å. However, RMSD profiles show the structural deviation of WT and G12C below 2Å.

We thank the reviewer for commenting on this apparent discrepancy. The two RMSD values were calculated for different parts of RAS. RMSD profiles show the overall structural deviation of WT and G12C below 2Å, which refer to the whole Ras protein structure (residue 1-166), calculated by the C_α atoms. This indicates that the overall RAS structure is stable in MD simulations. Then, from B-factor analysis, it shows that Switch I (corresponds to residue 25-40) and Switch II (corresponds to residue 58-75) and C-terminal are most dynamic regions in RAS. Next, MD snapshots are clustered and compared with available crystalized GTP-bound HRAS by RMSD analysis of Switch I and II backbone (CA, N, C and O atoms). The RMSDs of cluster-3 with regard to WT and G12C are only for the most dynamic region (Switch I and II), and thus it is reasonable that it has a higher RMSD than the overall structure of Ca atoms. In our revised manuscript, we changed caption from “*RMSD profile evolution in MD simulations*” to “*RMSD profile for the whole RAS structure (Ca atoms) in MD simulations*”, to make it clearer.

10. In Extended Data Figure 10 Panel C, more clarification/additional labeling is required. Are the first grouping of 3 timepoints for KRAS WT in RASless MEFs or KRAS G12C? With the current labeling here (and in general for Extended Figure 10/11), it is unclear how some of these

experiments differ in design from those in the main text and why some trends (RAS levels, pERK/ERK) are different with some of the supplemental data.

We thank the reviewer for pointing out missing labels and a lack of clarity of this figure. We have revised this figure with additional labels (see below). To address the reviewer's second point regarding redundancy in the experimental design. We agree with the reviewer that some data may be redundant, for example, a part of Extended Data Fig.10d overlaps with Fig. 4b, where we demonstrated the efficacy of VHL-Mb in G12V cell line. Also, a part of Fig. 4b that illustrates the efficacy of VHL-MB in G12C cell line overlaps with Fig. 4c. We have removed these redundancies in the figures. However, some experimental designs do have subtle differences and implications. For instance, the experimental design of Supplementary Fig. 11a is different from that of Fig. 4c. The purpose of Supplementary Fig. 11a is to show the cells at the time of implantation are still capable of degrading KRAS and inhibiting pERK upon dox induction at a time point close to implantation after expansion in flasks in order to perform the xenograft experiment (from a few million cells to ≥ 100 million cells), i.e. to exclude the possibility that the expanded cells had lost degrader expression due to subtle selection pressure. We have revised the legend to clearly state the motivation of the experiments in this figure.

Supplementary Fig. 10c

Reviewer #3 (Remarks to the Author):

This work builds on previous efforts by the authors to discover monobodies that can interact with Ras. The work is sufficiently novel to warrant publication in Nature Communications. Specifically, the discovery biochemical, structural and functional characterization of small proteins that can discriminate between WT and mutant Ras to block signaling and the complementary idea of coupling these selective agents to E3 ligases (VHL in this case) to trigger mutant-specific degradation are interesting and timely contributions to the field. In addition, the manuscript is well-written and the experiments appear carefully done. I have a few minor queries or suggestions that the authors might want to consider more as food for thought rather than as required changes prior to publication.

We are pleased that the reviewer recognized the novelty and timely contributions of our work. We also thank the reviewer for carefully reviewing the manuscript and for her/his constructive suggestions.

1) In Figure 1d (and, in general many of the western blots), it is confusing as to which tag was used for the IP versus the western blot. It would be easier for the reader if this information was added to the figure.

We thank the reviewer for pointing this out. We have made improvement to how we describe which tag was used for IP versus the whole cell lysate on our western blot involving pull-downs in the revised manuscript.

2) Please add the resolution of the structures either to the main methods or to the figure legend.

We have added the resolution of the crystal structure to the figure legends and also in the main text.

3) Line 106: Consider explaining that monobodies are FnIII-derived scaffolds for readers that are less familiar with the authors' previous work.

We thank the reviewer for this suggestion. We have included more background on the monobody technology under Introduction (L. 69-79).

4) Line 92: Consider describing this work as the “discovery” of a noncovalent inhibitor of Ras rather than “development” which, at least for some scientists, implies clinical evaluation of a potential therapeutic.

We thank the reviewer for this suggestion. However, because this molecule is a fruit of extensive engineering and not a naturally occurring compound and because there are numerous design aspects in addition to library screening, we feel that it is more appropriate to call the monobody generation process “development”. As such, we would like to retain the term, development. However, we accept the reviewer’s point in that we should differentiate the monobody from therapeutics until we have the results to demonstrate its therapeutic efficacy.

5) Line 152 Consider explaining what “computational structural analysis” was used to reveal the shallow pocket (visual inspection in PyMol/Coot? Or a specific program?) More generally, the authors rule out a change in backbone conformation being the cause of 12VC1 mutant selectivity. How much backbone change is there between the wild type and mutant forms of Ras in the absence of monobodies?

Thank you for your helpful comments. We employed AlphaSpace, a fragment-centric topographical mapping (FCTM) tool (J Chem Inf Model. 2015, 55: 1585), to detect targetable pockets in PPI interfaces. This tool has been successfully employed to design novel PPI inhibitors (J. Am. Chem. Soc. 2017, 139, 44, 15560; Nat. Commun., 11, 1786 (2020).). In current study, we detected binding pockets in the concave surface of monobody 12VC1 using AlphaSpace. To clarify this point, we have modified the text to the following and add the reference, “we utilized computational structural analysis AlphaSpace²⁸...”

28. D. W. Rooklin, C. Wang, J. Katigbak, P. S. Arora, and Y. Zhang, J. Chem. Inf. Model. , 55 , 1585 - 1599 (2015). AlphaSpace: Fragment-Centric Topographical Mapping to Target Protein-Protein Interaction Interfaces

Regarding the reviewer's second question, we have performed MD simulations for two crystalized RAS structures (WT and G12C) in the absence of monobodies (see Supplementary Fig.6a). For each starting conformation, we have performed 2 independent MD runs. The RMSD profiles, which calculated by the C_alpha atoms, show no big change (predominantly between 0.5 to 1.5 Å) in absence of monobodies. In addition, we also clustered these MD snapshots, and compared these MD clusters with the available crystalized GTP-bound HRAS as well as our two conformations ("RMSD analysis of Switch I and II backbone"). The results showed that even for the most dynamic Switch I and II, no obvious backbone changes (the RMSD values for each cluster are all within 1.0 Å between WT and G12C) was observed between WT and G12C in the absence of monobodies. We have now included detailed descriptions of computational analysis in the Methods section.

6) Figure 4b: Quantification of how much mutant Ras (rather than total Ras) is degraded by the monobody-VHL fusion would strengthen this section. More generally, while the monobody-VHL fusion is very clever and, as the authors point out, is potentially a great biological tool to more generally evaluate the potential of "degraders" against a given target without having to invest in the discovery of small molecule ligands, it isn't clear from comparison of the tumor efficacy data in figures 3 and figure 4 whether the monobody-VHL fusion has any greater effect than the inhibitory monobody itself in this case.

We thank the reviewer for this suggestion. Unfortunately, there's a lack of specific mutant-selective antibody for detecting the presence of RAS mutant in immunoblotting. We have reblotted some data with a KRAS-specific antibody to strengthen this section, mainly for G12V cell lines (see our response to Reviewer 1, point 5).

Indeed, it is not apparent whether inhibition or degradation would be more beneficial in the case of reducing tumor burden based on the available data. This is a great advice for future work. We have stated this point in the Discussion section of the revised manuscript (L. 364-365). We strongly agree with the reviewer that the monobody technology is great for generating tool biologic for investigating whether degradation or inhibition is more beneficial for a specific biological target. It will certainly be an important topic of our future studies.

7) Extended data figure 10: In Figure 10c, Ras degradation seems less profound that in Figure 4b. What is the difference between these experiments? Presumably in the RasLess MEFs, there is only mutant Ras? Did the authors measure the on and off rates of 12VC1.1 and 12VC1.2? Could this possibly explain the differences in efficacy?

We thank the reviewer for this comment. This confusion is entirely due to missing labels, as pointed out by the other reviewers. We have fixed this mistake in the revised manuscript (see below). The last lane of this panel, where we can see RAS level reduce from 1.0 to 0.2, is the only configuration of the PROTAC that functioned well for degrading RAS G12C. This figure demonstrates that although 12VC1.1 has higher affinity, 12VC1.2 is more efficient at degrading RAS. Although we have not measured the binding kinetic of 12VC1.2 and 12VC1.1, yeast display binding measurement revealed that 12VC1.2 has a weaker affinity than 12VC1.1. Affinity of a simple protein-protein interaction, e.g. monobody-RAS interaction, is generally well correlated with the dissociation rate, and thus we agree with the reviewer's expectation that

12VC1.2 has a faster off rate than 12VC1.1, which could rationalize the difference in degradation efficiency. We have noted this point (L. 267-273).

Supplementary Fig. 10c

8) Figure 3d – why is pERK down so much in all tumor samples, even when most of them don't show monobody expression (at least in the surviving tumor cells at the end of the experiment)? pERK levels in this blot aren't addressed in the text.

We appreciate the reviewer for pointing this out. Please see our response to Reviewer #2 Point 6.

Editorial Requests

Please deposit all proteomics raw data to a ProteomeXchange member repository and provide the accession codes as well as reviewer login information (see for example <https://www.ebi.ac.uk/pride/help/archive/reviewers>) in the Data Availability statement .

We have deposited the proteomics raw data to the MassIVE database. The accession code is provided under the Data Availability statement. The dataset is already publicly available.

When describing the LC-MS experiments, please state the total number of samples analyzed, numbers and types of controls, number of technical and/or biological replicates (even if n=1)

Please describe all relevant parameters of the LC-MS experiment (particularly LC gradient, gas phase fragmentation settings, mass resolution of the MS1 and MS2 scans, etc)

Please provide a full description of the database search parameters and acceptance criteria used for peptide identification (name and version of identification software, name and version of protein sequence database, protease cleavage sites and number of missed cleavages, fixed and variable modifications, mass tolerance for precursor and fragment ions, any applied score cutoffs, peptide- and protein-level FDR, minimum number of unique peptides for protein identification)

The information has been added to Supplementary Method 1.

In the legend of Extended Data Fig 3b, you mention that the listed listed proteins were uniquely captured from the lysate of PATU8902. If so, how were you then able to calculate a ratio of spectral counts recorded from PATU8902 lysate over the spectral counts recorded from the control (A375) lysate?

The "ratio" was calculated by the SAINT program (ref #50). It is correct that one cannot determine the true ratio if the denominator is zero. To avoid this problem, the program adds a value of 0.1 to all spectral counts prior to calculating the ratios. Therefore, as pointed out, the resulting numbers are not true ratios. Consequently, we have renamed them as "enrichment scores" and described how they are derived in Fig. 1 legend and in Methods.

NCOMMS-20-42599A Teng et al.
Responses to reviewers' comments

We are pleased that the revised manuscript was able to fully satisfy reviewers #1 and #3, and addressed a majority of the key concerns of reviewer #2. Once again, we would like to thank the reviewers for taking the time to carefully review our manuscript.

Regarding the western blot data shown in Figure 3d, the reviewers highlighted two concerns with this blot, (i) the unequal loading of the samples, and (ii) the pERK levels of the tumors are lower than that of the parental cell lines. Below we will address these two points separately.

To address the issue of uneven loading, we have repeated the acquisition of this particular data set. In this repeat we adjusted the amounts of the cell lysates to be loaded based on the total ERK levels, in order to make the loading more equal across the lanes. The repeated experiment yielded results similar to the original (see the images below). The samples (tumor ID# 1b, 1c, 2c) that have barely visible ERK had reached their maximum loading volume of our SDS gel system. The low protein concentrations of these samples reflect the sizes of the tumors shown in Fig3C and Supplementary Figure 8C, where some tumors were very small. Even though we adjusted the volume of the lysis buffer to the weight of each tumor, the small tumors still yielded lower protein concentrations. Therefore, we conclude that the issue of uneven loading was caused by the samples themselves and not by a technical error.

To address the issue that the pERK levels of the tumors are lower than that of the parental cell lines, we would like to emphasize that the cells for the “pre-injected cells (P)” lane were cultured on 2D surfaces (plastic dishes), whereas the tumor cells were grown in 3D inside mice. We do not expect the pERK levels from these distinct samples to be similar. Indeed, pERK levels from cells cultured in 2D versus 3D samples have been shown to be significantly different when directly compared to each other^{1,2}. Furthermore, tumor samples are heterogenous, containing other cells in addition to the tumor cells. These key physiological differences may well attribute to the different pERK levels observed in this experiment. As such we cannot draw a conclusion regarding the state of the tumor cells by comparing their pERK levels with that of the pre-injected cells. We regret that we did not make this point clear in our response.

The key takeaways from this figure, as we have highlighted in several places in the rebuttal and in the main manuscript, are (i) that the tumor cells at the end of xenograft experiment no longer express the mutant selective inhibitory monoclonal antibody (12VC1) (see the blots with the red arrows), (ii) that the negative control monoclonal antibody (Mb(Neg)) is still robustly expressed. We would also like to emphasize that pERK levels in these samples at the end of the experiment do not inform the efficacy of the inhibitory monoclonal antibody, because the expression of the monoclonal antibody has been lost from these cells that survived.

Again, we do not expect that the pERK levels in the recovered tumors should be similar to those in the parental cells grown in dishes. These pERK blots were included primarily for the sake of completeness. Similarly, in the original and repeated data, the pERK levels from the tumors were about half of the parental lines. The general concordance

between the two experiments suggests that the differences in the pERK level reflect inherent differences across samples, rather than technical errors. We will include descriptions explaining our finding regarding variations in the pERK level across the tumor samples compared with the parental cell lines in the main text.

The red arrows point to the expression levels of the monobodies at the end of the xenograft experiment.

To direct the reader to focus on the salient points in this figure, we will rearrange the panels so that the blot for monobody expression is at the top.

To address the rest of the reviewer's discussion point #6, which stated that proliferated tumor cells should have higher pERK levels than the rest of the samples, we argue that this is not necessarily the case. Tumor cells that has lost monobody expression should have a pERK level similar to that in the un-induced tumor cells and also to that in cells that expresses non-binding monobody.

We will be happy to address the other comments by Reviewer #2.

We hope that these explanations and the additional data address the remaining concerns.

References

1. Breslin S, O'Driscoll L. The relevance of using 3D cell cultures, in addition to 2D monolayer cultures, when evaluating breast cancer drug sensitivity and resistance. *Oncotarget*. 2016;7(29):45745-45756. doi:10.18632/oncotarget.9935
2. Jung HR, Kang HM, Ryu JW, Kim DS, Noh KH, Kim ES, Lee HJ, Chung KS, Cho HS, Kim NS, Im DS, Lim JH, Jung CR. Cell Spheroids with Enhanced Aggressiveness to Mimic Human Liver Cancer In Vitro and In Vivo. *Sci Rep*. 2017 Sep 5;7(1):10499. doi: 10.1038/s41598-017-10828-7. PMID: 28874716; PMCID: PMC5585316.

REVIEWERS' COMMENTS

Reviewer #1 (Remarks to the Author):

The authors have addressed all of the concerns mentioned in the initial review of the paper. Inclusion of the experimental scheme in Fig4a and Supplementary Fig10e make it much easier to understand the experimental setup. The addition of biological replicates as well as UPS inhibitor experiments for the G12V monobody degraders strength the presented data. The fluorescence intensity profile in figure 1c makes it much easier to visualize the co-staining of the monobody and GFP-KRAS. Changes to tone and the brief mention of limitations is appreciated and does not detract from the significance or novelty of the study. Finally, grammatical and stylistic edits make the manuscript easier to read. Due to the thoroughness in addressing all concerns raised during the initial review, I support publication of this study.

Reviewer #2 (Remarks to the Author):

The investigators have now addressed a number of concerns in the revised manuscript, including issues with technical variability/limited impact on endogenous KRAS degradation. There is also notable improvement in the clarity between similar experiments in the main/supplemental figures. However, a few concerns still remain.

1. Discussion point 3: Overall, the discussion of the discrepancy between G12S and G12C affinity is sufficient. However, the investigators' response is partly addressed in the text and the rebuttal. This discussion should be consolidated and expanded within the text (page 8, lines 170-179).

2. Discussion point 4: In Figure 2 legend, the structure was solved in the GTP-bound form but the nucleotide-bound state is not mentioned." This clarification was not addressed by the authors.

3. Discussion point 6. The additional normalization of pERK to total ERK levels is very useful. However, the original concern has not been fully addressed. Though the authors state that pERK levels are not dramatically decreased, the quantification shows a similar 2-fold decrease across all samples, despite the fact that there are tumor size differences. One would expect that the 12VC1-dox samples that show significantly larger tumor sizes due to "unimpeded" proliferation would show

actual increases in pERK signaling given the KRAS mutation. Instead, signaling looks identical to the other 3 conditions.

4. Discussion point 5. The additional discussion of variability in tumor injection, etc. as well as proof of concept for stability of expression in the 293T/A375 lines is very useful. This in combination with findings that the xenograft experiments in the H23 cell line did not show differences between -dox MB(neg) and 12VC1, alleviate some of the concerns. However, along the lines of comments regarding discussion point 6, quantification of blots for this experiment set (supplemental figure 11) show significant differences in pERK between treatment and negative controls. In addition, Fig. 3 uses just the monobody, while Fig. 4/supplemental Fig. 11 uses monobody+VHL for degradation, so increased efficacy is not unexpected. With this in mind, some discussion is needed regarding findings that pERK doesn't change in the PATU8902 xenografts in Fig. 3.

5. It is unclear why the y-axis of the right panel of Supplemental Figure 1a is 60000, while it is 30000 in the left panel. To better compare these panels, they should be scaled similarly.

Reviewer #3 (Remarks to the Author):

The authors have addressed my concerns. It would be nice if the loading controls on figure 3d were more consistent which would make interpretation of the pERK levels easier. Having said that, the rest of the results are consistent with the interpretation.

NCOMMS-20-42599A
Responses to the reviewers' comments

Responses are in blue.

Reviewer #1 (Remarks to the Author):

The authors have addressed all of the concerns mentioned in the initial review of the paper. Inclusion of the experimental scheme in Fig4a and Supplementary Fig10e make it much easier to understand the experimental setup. The addition of biological replicates as well as UPS inhibitor experiments for the G12V monobody degraders strength the presented data. The fluorescence intensity profile in figure 1c makes it much easier to visualize the co-staining of the monobody and GFP-KRAS. Changes to tone and the brief mention of limitations is appreciated and does not detract from the significance or novelty of the study. Finally, grammatical and stylistic edits make the manuscript easier to read. Due to the thoroughness in addressing all concerns raised during the initial review, I support publication of this study.

We are pleased that we have been able to fully address this reviewer's concerns.

Reviewer #2 (Remarks to the Author):

The investigators have now addressed a number of concerns in the revised manuscript, including issues with technical variability/limited impact on endogenous KRAS degradation. There is also notable improvement in the clarity between similar experiments in the main/supplemental figures. However, a few concerns still remain.

1. Discussion point 3: Overall, the discussion of the discrepancy between G12S and G12C affinity is sufficient. However, the investigators' response is partly addressed in the text and the rebuttal. This discussion should be consolidated and expanded within the text (page 8, lines 170-179).

We have consolidated the responses in the main text (lines 168-186). Please note that the line numbers in the marked version are incorrect due to a mysterious error in MS Word. The program skips line numbers across pages in an unpredictable manner. The line numbers in this document refer to those in the clean version. Apologies for the inconvenience.

2. Discussion point 4: In Figure 2 legend, the structure was solved in the GTP-bound form but the nucleotide-bound state is not mentioned." This clarification was not addressed by the authors.

Thank you for pointing out this omission. "bound to GTP γ S" has been added to Figure 2 legend (line 1006).

3. Discussion point 6. The additional normalization of pERK to total ERK levels is very useful. However, the original concern has not been fully addressed. Though the authors state that pERK levels are not dramatically decreased, the quantification shows a similar 2-fold decrease across all samples, despite the fact that there are tumor size differences. One would expect that the 12VC1-dox samples that show significantly larger tumor sizes due to "unimpeded" proliferation would show actual increases in pERK signaling given the KRAS mutation. Instead, signaling looks identical to the other 3 conditions.

4. Discussion point 5. The additional discussion of variability in tumor injection, etc. as well as proof of concept for stability of expression in the 293T/A375 lines is very useful. This in combination with findings that the xenograft experiments in the H23 cell line did not show differences between -dox MB(neg) and 12VC1, alleviate some of the concerns. However, along the lines of comments regarding discussion point 6, quantification of blots for this experiment set (supplemental figure 11) show significant differences in pERK between treatment and negative controls. In addition, Fig. 3 uses just the monobody, while Fig. 4/supplemental Fig. 11 uses monobody+VHL for degradation, so increased efficacy is not unexpected. With this in mind, some discussion is needed regarding findings that pERK doesn't change in the PATU8902 xenografts in Fig. 3.

Comments 3 and 4, as well as the comment by Reviewer 3, are on the same point. Thus, we address them together.

Regarding the western blot data shown in Figure 3d, the reviewers highlighted two concerns with this blot, (i) the unequal loading of the samples, and (ii) the pERK levels of the tumors are lower than that of the parental cell lines. Below we will address these two points separately.

To address the issue of uneven loading, we have repeated the acquisition of this particular data set. In this repeat we adjusted the amounts of the cell lysates to be loaded based on the total ERK levels, in order to make the loading more equal across the lanes. The repeated experiment yielded results similar to the original (see the images below). The samples (tumor ID# 1b, 1c, 2c) that have barely visible ERK had reached their maximum loading volume of our SDS gel system. The low protein concentrations of these samples reflect the sizes of the tumors shown in Fig3C and Supplementary Figure 8C, where some tumors were very small. Even though we adjusted the volume of the lysis buffer to the weight of each tumor, the small tumors still yielded lower protein concentrations. Therefore, we conclude that the issue of uneven loading was caused by the samples themselves and not by a technical error.

To address the issue that the pERK levels of the tumors are lower than that of the parental cell lines, we would like to emphasize that the cells for the “pre-injected cells (P)” lane were cultured on 2D surfaces (plastic dishes), whereas the tumor cells were grown in 3D inside mice. We do not expect the pERK levels from these distinct samples to be similar. Indeed, pERK levels from cells cultured in 2D versus 3D samples have been shown to be significantly different when directly compared to each other^{1,2}. Furthermore, tumor samples are heterogenous, containing other cells in addition to the tumor cells. These key physiological differences may well attribute to the different pERK levels observed in this experiment. As such we cannot draw a conclusion regarding the state of the tumor cells by comparing their pERK levels with that of the pre-injected cells. We regret that we did not make this point clear in our response.

The key takeaways from this figure, as we have highlighted in several places in the rebuttal and in the main manuscript, are (i) that the tumor cells at the end of xenograft experiment no longer express the mutant selective inhibitory monobody (12VC1) (see the blots with the red arrows), (ii) that the negative control monobody (Mb(Neg)) is still robustly expressed. We would also like to emphasize that pERK levels in these samples at the end of the experiment do not inform the efficacy of the inhibitory monobody, because the expression of the monobody has been lost from these cells that survived.

Again, we do not expect that the pERK levels in the recovered tumors should be similar to those in the parental cells grown in dishes. These pERK blots were included primarily for the sake of completeness. Similarly, in the original and repeated data, the pERK levels from the tumors

were about half of the parental lines. The general concordance between the two experiments suggests that the differences in the pERK level reflect inherent differences across samples, rather than technical errors. We will include descriptions explaining our finding regarding variations in the pERK level across the tumor samples compared with the parental cell lines in the main text.

The red arrows point to the expression levels of the monobodies at the end of the xenograft experiment.

To direct the reader to focus on the salient points in this figure, we will rearrange the panels so that the blot for monobody expression is at the top.

To address the rest of the reviewer's discussion point #6, which stated that proliferated tumor cells should have higher pERK levels than the rest of the samples, we argue that this is not necessarily the case. Tumor cells that has lost monobody expression should have a pERK level similar to that in the un-induced tumor cells and also to that in cells that expresses non-binding monobody.

References

1. Breslin S, O'Driscoll L. The relevance of using 3D cell cultures, in addition to 2D monolayer cultures, when evaluating breast cancer drug sensitivity and resistance. *Oncotarget*. 2016;7(29):45745-45756. doi:10.18632/oncotarget.9935
2. Jung HR, Kang HM, Ryu JW, Kim DS, Noh KH, Kim ES, Lee HJ, Chung KS, Cho HS, Kim NS, Im DS, Lim JH, Jung CR. Cell Spheroids with Enhanced Aggressiveness to Mimic Human Liver Cancer In Vitro and In Vivo. *Sci Rep*. 2017 Sep 5;7(1):10499. doi: 10.1038/s41598-017-10828-7. PMID: 28874716; PMCID: PMC5585316.

We have revised the relevant paragraph (l.249-266) and added a sentence in Figure 3 legend (l.1051-1058).

5. It is unclear why the y-axis of the right panel of Supplemental Figure 1a is 60000, while it is 30000 in the left panel. To better compare these panels, they should be scaled similarly.

Thank you for pointing out this discrepancy. The figures have been revised using the same vertical axis range. This change does not affect the interpretation or conclusion.

Reviewer #3 (Remarks to the Author):

The authors have addressed my concerns. It would be nice if the loading controls on figure 3d were more consistent which would make interpretation of the pERK levels easier. Having said that, the rest of the results are consistent with the interpretation.

Please see above the response to Reviewer #2's comments 3 and 4.